# In Vitro Evaluation of the Antiamoebic Activity of Kaempferol against Trophozoites of *Entamoeba histolytica* and in the Interactions of Amoebae with Hamster Neutrophils

**DOI:** 10.3390/ijms241311216

**Published:** 2023-07-07

**Authors:** David Levaro-Loquio, Jesús Serrano-Luna, Maritza Velásquez-Torres, Germán Higuera-Martínez, Ivonne Maciel Arciniega-Martínez, Aldo Arturo Reséndiz-Albor, Nadia Mabel Pérez-Vielma, Judith Pacheco-Yépez

**Affiliations:** 1Sección de Estudios de Postgrado, Escuela Superior de Medicina, Instituto Politécnico Nacional, Ciudad de México 11340, Mexico; dlevarol1600@alumno.ipn.mx (D.L.-L.); mvelasquezt1500@alumno.ipn.mx (M.V.-T.); ghigueram1700@alumno.ipn.mx (G.H.-M.); iarciniega@ipn.mx (I.M.A.-M.); aresendiza@ipn.mx (A.A.R.-A.); 2Departamento de Biología Celular, Centro de Investigación y de Estudios Avanzados del IPN (CINVESTAV), Av. IPN No. 2508 Col. San Pedro Zacatenco, Ciudad de México 07360, Mexico; jesus.serrano@cinvestav.mx; 3Centro Interdisciplinario de Ciencias de la Salud, CICS, San Tomás, Instituto Politécnico Nacional, Ciudad de México 11340, Mexico; nperezv@ipn.mx

**Keywords:** *Entamoeba histolytica*, neutrophil, kaempferol, metronidazole, hamster, myeloperoxidase, reactive oxygen species, nitric oxide, thioredoxin reductase, thioredoxin, peroxiredoxin, rubrerythrin

## Abstract

*Entamoeba histolytica* (*E. histolytica*) is a parasite in humans that provokes amoebiasis. The most employed drug is metronidazole (MTZ); however, some studies have reported that this drug induces genotoxic effects. Therefore, it is necessary to explore new compounds without toxicity that can eliminate *E. histolytica*. Flavonoids are polyphenolic compounds that have demonstrated inhibition of growth and dysregulation of amoebic proteins. Despite the knowledge acquired to date, action mechanisms are not completely understood. The present work evaluates the effect of kaempferol against *E. histolytica* trophozoites and in the interactions with neutrophils from hamster, which is a susceptibility model. Our study demonstrated a significant reduction in the amoebic viability of trophozoites incubated with kaempferol at 150 μM for 90 min. The gene expression analysis showed a significant downregulation of Pr (peroxiredoxin), Rr (rubrerythrin), and TrxR (thioredoxin reductase). In interactions with amoebae and neutrophils for short times, we observed a reduction in ROS (reactive oxygen species), NO (nitric oxide), and MPO (myeloperoxidase) neutrophil activities. In conclusion, we confirmed that kaempferol is an effective drug against *E. histolytica* through the decrease in *E. histolytica* antioxidant enzyme expression and a regulator of several neutrophil mechanisms, such as MPO activity and the regulation of ROS and NO.

## 1. Introduction

*E. histolytica* is a protozoan pathogen in humans and is the causative agent of amoebiasis. Between 40 and 50 million people worldwide are infected with this protozoan parasite, representing a major public health problem. Approximately 100,000 patients die each year worldwide [1]. *E. histolytica* has been reported to have a high tissue-destroying capacity. According to the World Health Organization (WHO), amoebiasis is the third leading cause of death due to parasitic disease. In Mexico, amoebiasis has a considerable prevalence in both cities and rural areas; the prevalence of this parasite is highest in the states of Morelos (19%) and Chiapas (51%) [2]. Amoebiasis is a gastrointestinal infection that can cause amoebic colitis. The most common symptoms are diarrhea with blood and mucus, and various alterations may occur depending on the location. The most common extraintestinal form is the amoebic liver abscess (ALA), which occurs when amoebae are transported through the bloodstream from the intestine to the liver. *E. histolytica* in the liver can induce the formation of lesions in the hepatic parenchyma, presenting an important inflammatory response, which is constituted by polymorphonuclear leukocytes (PMNs), mainly neutrophils (acute phase), as well as by mononuclear cells such as macrophages (chronic phase). The sequence of events during ALA formation was described in the susceptible model, hamster (*Mesocricetus auratus*) [3].

Previous studies reported that in hamster (susceptible) and mouse (resistant) models of ALA, the first cells to come into contact with the amoebae are neutrophils, where MPO is the most important enzyme secreted in the azurophilic granules of neutrophils and monocytes [4]. In in situ assays, the hamster model showed lower expression and activity of this enzyme than the mouse model [5]. The substrate of MPO is hydrogen peroxide (H_2_O_2_), which results from the reaction between molecular oxygen and the NADPH oxidase complex [6]; MPO uses H_2_O_2_ to catalyze the oxidation of two chloride ions (Cl-) to produce hypochlorous acid (HOCl) [7]. HOCl is an oxidizing agent that causes damage to DNA, lipids, and proteins [8]. Several studies have demonstrated MPO activity in chronic inflammatory diseases such as cystic fibrosis and other autoimmune diseases [9,10].

*E. histolytica* has developed highly efficient enzyme systems to resist oxidative damage from ROS and, thus, maintain the intracellular redox balance. One of these systems is the thioredoxin system, which is composed of NADPH, TrxR, Trx (thioredoxin), and Prx and constitutes a key antioxidant system in defense against oxidative stress. Additionally, TrxR catalyzes the reversible transfer of reducing equivalents between NADPH and Trx, a small protein that plays a key metabolic function in maintaining the intracellular redox balance [11]. TrxR on *E. histolytica* was found to be a nitroredoxin reductase capable of reducing nitroimidazoles, azomycin, and MTZ in vitro [12]. In addition, the main mechanism of peroxynitrite (ONOO^−^) interception is probably this system in conjunction with Prx [13]. Prx can also degrade H_2_O_2_ [14]. The Trx system in *E. histolytica* could also offer an alternative protective mechanism against lipid peroxidation, which would allow for intracellular proteins and DNA itself to be safe from highly toxic ROS and reactive nitrogen species (RNS). If so, this would represent a key mechanism in relation to the virulence of *E. histolytica* when exposed to highly toxic reactive oxygen intermediates [15]. Rr is a protein that has a pro-oxidase activity, as it is involved in the detoxification of H_2_O_2_ [16], and the enzyme can utilize both NADPH and NADH as reducing equivalent donors for reductase or oxidase activity [17].

Flavonoids are natural pigments present in vegetables, fruits, and certain beverages; they present several biochemical and antioxidant effects favorable to the resolution of various diseases such as cancer, Alzheimer’s disease, and atherosclerosis, among other ailments [18]. The general structure of flavonoids is characterized by the presence of a variable number of phenolic hydroxyl groups (polyphenols), which have a low molecular weight (500 to 4000 Da). Their structure is composed of a common skeleton of diphenylpyranes (C6-C3-C6), in addition to two phenyl rings linked through a C ring of pyrene. Flavonoids have been used to treat diseases caused by human parasites in traditional medicine practices in America, Africa, and Asia. Many of the major flavonoid groups have been shown to exhibit antiparasitic activity; this effect has been reported against the following parasites: *Plasmodium falciparum*, *Trypanosoma brucei brucei*, *Trypanosoma brucei gambiens* and *Trypanosoma cruzi*, *Leishmania donovani*, *Cryptosporidium parvum*, and *Toxoplasma gondii* [19].

Among these flavonoids, kaempferol (3,5,7-trihydroxy-2-(4-hydroxyphenyl)-4H-1-benzopyran-4-one) is a compound of a low molecular weight (MW: 286.2 g/mol) with the classic flavonoid diphenylpyran structure (C6-C3-C6). This molecule has been identified in many plant species commonly used in traditional medicine; therefore, it has been the subject of numerous studies. Kaempferol has a wide range of biological activities, such as antioxidant, anti-inflammatory, anticancer, antibacterial, and antiviral activities, as well as against several protozoa [20]. Moreover, kaempferol is an extremely active ROS scavenger [21]. Kaempferol has been reported to have an effect in suppressing ROS production in mouse bone marrow-derived neutrophils [22]. Moreover, in a model of induced mouse mastitis [23] and a mouse model of LPS-induced acute lung injury treated with kaempferol, the MPO activation was reduced [24].

Kaempferol was studied previously in the interaction with *E. histolytica* trophozoites, and the authors reported the deregulation of amoebic actin and myosin II heavy chain; cortexylin II, a heat shock protein; glyceraldehyde phosphate dehydrogenase; and fructose-1, 6-bisphosphate aldolase (G/FBPA). The authors also showed that at 48 h of kaempferol incubation a concentration of 27.7 μM was able to inhibit 77.1% of amoebic growth compared with trophozoites not treated with this flavonoid [25]. These results are similar to the results reported with (-)-epicatechin [26]. Previous studies analyzed a slight ROS production in 5 × 10^5^ *E. histolytica* trophozoites incubated with kaempferol at 27.7 μM for 48 h compared with trophozoites without kaempferol [27].

In the present work, we analyze the effect of kaempferol on *E. histolytica* and in the interaction of *amoebae* with hamster neutrophils for short times, demonstrating that kaempferol has a dual effect on amoeba and neutrophils. On the one hand, it diminished amoebic viability; on the other hand, kaempferol also decreased neutrophils’ MPO activity and ROS and NO release. In a hamster susceptible model to ALA, kaempferol could participate in amoebic liver abscess resolution through its anti-inflammatory and antioxidant activities.

## 2. Results

### 2.1. Effect of Kaempferol on the Viability of E. histolytica and VERO Cells

To analyze the effect of kaempferol on the viability of *E. histolytica*, WST-1 reagent was used. For this assay 15,000 *E. histolytica* trophozoites were incubated with concentrations of 90, 100, 110, 120, 130, 140, and 150 μM kaempferol or MTZ at 90 min. When analyzing the viability of the trophozoites incubated at concentrations of 90, 100, and 110 μM of kaempferol and MTZ, we did not observe a decrease in the viability of the trophozoites, but when kaempferol was used at concentrations of 130, 140, and 150 μM, we observed a viability of 44.5% with kaempferol at 150 μM. We also observed a significant difference in the viability of kaempferol compared with MTZ at these concentrations (Figure 1).

When we evaluated the VERO cell viability at different times with kaempferol or MTZ at 150 μM using the WST-1 assay, we did not observe any significant differences at any of the assayed times (Figure 2).

### 2.2. Proteins’ Overexpression of E. histolytica in the Presence of Kaempferol

After obtaining the total cell lysates of *E. histolytica* in the presence of kaempferol or MTZ, a protein separation analysis was performed in a 10% SDS-polyacrylamide gel and detected by staining with Coomassie Blue (Figure 3). When analyzing trophozoites incubated with 150 μM kaempferol for 90 min, we observed an overexpression of molecular bands of approximately 67, 59, 27, and 24 kDa (C). On the other hand, we observed a difference in the 67 kDa band in the MTZ samples (B) compared with the Ctrl (A). 

### 2.3. Downregulation of E. histolytica-Detoxifying Enzyme Expression in the Presence of Kaempferol

Samples of cDNA obtained from trophozoites of *E. histolytica* in the presence of kaempferol or MTZ at 150 μM for 90 min were analyzed using qRT-PCR for the quantification of the detoxifying enzyme gene expression (Figure 4). Using real-time PCR, we found a downregulation of three genes of trophozoites in the presence of kaempferol, Rr, which had a significant decrease of 84.04-fold compared with the basal control. In the case of Prx, we also observed a decrease of 58.49-fold against the basal control and a decrease of 3.97-fold of TrxR against the basal control. In contrast, we did not observe any difference in the gene expression of Trx in the kaempferol group compared with the basal control. On the other hand, we observed a downregulation of all genes in the kaempferol group compared with the MTZ group.

### 2.4. Kaempferol Induces a Decrease in MPO Activity in E. histolytica and Neutrophil Interactions

Supernatants of the interaction of trophozoites of *E. histolytica* incubated with hamster neutrophils in the presence of 150 μM kaempferol (N+T+kaempferol) or MTZ (N+T+MTZ) at different times (20, 40, 60, and 90 min) were collected (Figure 5). The results showed that neutrophil MPO activity decreased significantly in the presence of 150 μM kaempferol at 60 and 90 min of incubation compared with that of the respective control, neutrophils and trophozoites (N+T). We did not observe a significant difference in the DMSO group (N+T+DMSO) compared with the control (N+T); in contrast, when we compared it with the MTZ group (N+T+MTZ), we observed a significant increase at 90 min compared with N+T+kaempferol. We also provided a negative control with an inhibitor of MPO, ABAH (4-amino-benzoic acid hydrazide) (N+T+ABAH). 

### 2.5. Kaempferol Induces Low ROS Release in E. histolytica and Neutrophil Interactions 

The quantification of ROS from the interaction of *E. histolytica* with neutrophils showed that under basal conditions neutrophils and trophozoites produce a scarce amount of ROS (Figure 6a). As a positive control, we showed a group with LPS (lipopolysaccharide). In this group, we observed an increase in ROS activity compared with neutrophils alone. It is worth mentioning that when neutrophils interacted with the amoebae, an increase in ROS production was observed. In the presence of 150 μM of kaempferol at 90 min with trophozoites and neutrophils, we observed a significant decrease in ROS activity compared with the group in the absence of this drug, and we observed a significant difference compared with the MTZ group.

### 2.6. Kaempferol Induces a Decrease in NO Production in the E. histolytica and Neutrophil Interaction 

In the interaction between *E. histolytica* and neutrophils, an increase in NO concentration was observed compared with neutrophils (Figure 6b). When kaempferol (150 μM) was incubated in the presence of amoebic trophozoites with neutrophils, we observed a significant decrease in NO production compared with the interactions in the absence of kaempferol. Moreover, we observed a significant difference compared with the MTZ group.

## 3. Discussion

The drug of choice for amoebiasis is MTZ; however, due to its side effects in long-term treatments, it is essential to look for alternative therapies using new compounds that do not present adverse effects for humans. Kaempferol is a flavonoid whose antiamoebic activity has been described previously in an in vitro assay [20]; however, its effect on *E. histolytica* trophozoites at short times is unknown. Therefore, it is necessary to test its efficacy. Kaempferol possesses a wide range of biological activities, such as antioxidant, anti-inflammatory, anticancer, antibacterial, and antiviral activities, and it also has effective activity against several protozoa [20].

In our study, the results of the WST-1 assay showed that the OD values, were directly proportional to the viability of 15,000 *E. histolytica* trophozoites, showing a decrease in their viability by increasing the concentration of kaempferol (130, 140, and 150 μM) at 90 min of incubation compared with MTZ at the same concentrations, considering that MTZ is the drug of choice against amoebiasis. Furthermore, a concentration of 150 μM of kaempferol significantly decreased viability. Additionally, we evaluated the effect of kaempferol at a concentration of 150 μM on the viability of 10,000 VERO cells using the WST assay at 60, 90, 180, and 360 min (Figure 2). In this experiment, we observed that kaempferol had no effect on the viability of mammalian cells at this concentration and at these times.

Kaempferol is obtained from *Helianthemum glomeratum*, an endemic medicinal herb used to treat diarrhea, abdominal pain, and dysentery in Mexico, and it shows activity against *E. histolytica* at a concentration of 27.7 μM of kaempferol for 48 h [28]. After obtaining total cell lysates of *E. histolytica*, protein separation analysis was performed on a 10% SDS-polyacrylamide gel, where we observed overexpression of 67, 59, 27, and 24 kDa protein bands. A previous work reported an overexpression effect of a cytoskeleton protein (actin) in *E. histolytica* trophozoites incubated with kaempferol at a concentration of 27.7 μM for 48 h [25]. Even though the induction of the protein bands of the weight of 67, 59, 27, and 24 kDa is clearly observed in the presence of kaempferol in the SDS-polyacrylamide gel, due to the limitations of this technique, it is necessary in future studies to carry out two-dimensional gel assays together with the mass spectrometry technique to be able to state with precision which are the amoebic proteins that are being induced by kaempferol.

We also analyzed the relative gene expression of *E. histolytica* antioxidant enzymes in the presence of kaempferol at a concentration of 150 μM for 90 min, and we found downregulation of genes encoding *E. histolytica*-detoxifying enzymes such as Rr, Prx, and TrxR compared with the basal control and Rr, Prx, TrxR, and TrxR compared with the MTZ group; these enzymes are responsible for H_2_O_2_ detoxification and defense of amoeba against oxidative stress [29]. On the other hand, we did not observe any significant change in genes encoding other enzymes such as Trx compared with the control group. Compared to the MTZ group, Rr, Prx, TrxR, and Trx were downregulated in the presence of kaempferol; these enzymes are also related to the antioxidant capacity of the parasite. The downregulation in the expression of these antioxidant enzymes by kaempferol suggests that the enzymatic activities were depleted due to their participation in the regulation of the oxidative environment. Despite the results achieved on amoebic antioxidant enzymes, it will be necessary to evaluate their distribution and activity in the future. In the case of MTZ, we observed a clear upregulation of these genes. In other studies, it was reported that in *E. histolytica* trophozoites that were treated with MTZ at a concentration of 50 micromolar for 1, 3, or 5 h, there was an overexpression in the expression of Trx, TRxR, and Prx genes [30].

Importantly, MPO is an enzyme released by neutrophils that binds to *E. histolytica* trophozoites, causing damage to them [31]. When we performed tests to quantify the MPO in hamster neutrophils and trophozoites with 150 μM of kaempferol at different times, our results showed a significant decrease in MPO activity at 60 and 90 min of incubation compared with the control and MTZ, and the presence of this enzyme was significantly decreased at 90 min. Previous reports showed that in interactions of trophozoites with mouse and hamster neutrophils, MPO activity in the hamster model was lower than in the mouse model [32]. In this study, we demonstrated that kaempferol decreases the MPO activity in hamster neutrophils, which may favor the ALA resolution in a susceptible model where a prolongated and exaggerated inflammatory response exacerbates the amoebic damage [3,32].

In a previous work using Surface Plasmon Resonance (SPR) binding analysis methodology to describe the binding of flavonoids, it was reported that MPO was immobilized on a CM5 sensor chip. Aliquots of 40 μL of kaempferol were injected into the flow cell and it was observed that kaempferol at 25 μM bound the MPO after 120 s. It exhibited significant affinity for MPO. These authors claimed that this affinity could inhibit the activity of this enzyme, which could explain the decrease in MPO activity and the role of kaempferol as an inhibitor of this enzyme in neutrophils [33].

Neutrophils have various mechanisms to damage *E. histolytica* trophozoites such as ROS and NO. Neutrophils require the respiratory burst to release ROS in the presence of oxygen. This production is catalyzed by means of NADPH oxidase, which, when it is activated, generates two molecules of superoxide anion [34] and NO is synthesized by neutrophil nitric oxide synthase (iNOS) [35]. We reported a significant decrease in ROS and NO concentrations in interactions of trophozoites with hamster neutrophils in the presence of kaempferol, compared with the control group and interactions in the presence of MTZ. Kaempferol was used to inhibit the formation of ROS in mouse bone marrow-derived neutrophils stimulated with PMA at 10 nmol/L for 30 min due to the antioxidant capacity of the flavonoid [22]. In another study in rats treated with 10 mg/kg/day kaempferol, tissue gingival fibroblasts were obtained and homogenized; this drug may exert a profound inhibitory effect against the iNOS [36]. In another study using a model of a biological oxygen monitor, it was reported that a concentration of 92 μM kaempferol inhibited NADPH oxidase activity by decreasing ROS and NO production in human neutrophil stimulated by PMA [37]. A previous report of the modified xanthine/luminol/xanthine oxidase assay was used to evaluate the superoxide scavenging effect of kaempferol, and it was observed that this flavonoid is a potent superoxide anion scavenger that decreases superoxide anion levels [38]. Superoxide is necessary for the normal production of most ROS and NO species involved in oxidative stress [39].

Since neutrophils contribute to host tissue damage in ALA through the massive lysing of these cells and the release of oxidative products (ROS, NO), the scavenger effect of kaempferol may promote the resolution of amoebic abscess. Future studies in vivo are needed to understand the mechanisms of the positive antiamoebic effect of kaempferol in interactions between neutrophils and *E. histolytica* trophozoites.

Thus, in this study we confirmed kaempferol as an effective drug against *E. histolytica*, which has a dual effect on trophozoites and hamster neutrophils, through a regulatory effect on MPO, ROS, and NO in these interactions at short times. In the hamster, a chronic susceptible model to ALA, the reduction in the neutrophils’ MPO activity as well as the reduction in ROS and NO production could be the mechanisms that could participate in the amoebic liver abscess resolution.

## 4. Materials and Methods 

### 4.1. Amoebic Cultures

*E. histolytica* trophozoites of the HM-1: IMSS strain were cultured axenically at 37 °C in Diamond’s trypticase yeast iron extract (TYI-S-33) culture medium supplemented with 10% bovine serum (Gibco, BRL, Grand Island, NY, USA). The trophozoites were cooled, detached, and harvested at the end of the logarithmic growth phase (48 h) by chilling to 4 °C. The trophozoites were concentrated via centrifugation at 300× *g* for 5 min and used immediately [40]. 

### 4.2. VERO Cell Culture

VERO cell line (kidney epithelial cell derived from the African green monkey) (*Ceropithecus aethiops*) was obtained from the American Type Culture Collection (ATCC-CRL 1586). The cells were cultivated in DMEM/F-12/Media (Gibco, BRL, Grand Island, NY, USA) and 10% Fetal Bovine Serum (FBS) (Gibco, BRL, USA) and 1% antibiotic with 10,000 units penicillin and 10 mg streptomycin/mL (Sigma-Aldrich, St. Louis, MO, USA) at 37 °C in a 5% CO_2_ humidified atmosphere [41]. 

### 4.3. E. histolytica Viability Assays (WST-1)

WST-1 (2-(4-Iodophenyl)-3-(4-nitrophenyl)-5-(2,4-disulphophenyl)-2H-tetrazolium), a tetrazolium salt (Roche, IND, Basel, Switzerland), was used to evaluate the effect of the viability of kaempferol (Sigma-Aldrich, MO, USA) on trophozoites of *E. histolytica* [42,43]. Kaempferol was dissolved in 3 μL of DMSO (Sigma-Aldrich, MO, USA) and 997 μL of TYI-S-33 medium under sterile conditions to obtain a final concentration of 1 μg/μL. A total of 15,000 trophozoites were incubated in a 96-well plate at a final volume of 200 μL of sterile TYI-S-33 medium for 90 min in the presence of 90, 100, 110, 120, 130, 140, and 150 μM kaempferol and MTZ (FLAGYL, Sanofi, México). After the incubation time, the supernatant was removed and then washed 3 times with PBS at the same time the reagent WST-1 was prepared, taking 95 μL PBS + 5 μL of WST-1 and adding 100 μL of WST-1 to each well. The plates were then incubated and protected from light for 30 min at 37 °C. Then, 50 μL of the supernatant was taken from each well, placed in a new 96-well plate (Costar, New York, NY, USA), and finally read in a Synergy HT plate reader (Biotek, VER, Winooski, VT, USA) at a wavelength of 450 nm. Trophozoites were incubated in a complete TYI-S-33 medium and 0.2% DMSO was used a negative control in this experiment. Data are expressed as the mean ± standard deviation.

### 4.4. VERO Cell WST-1 Cytotoxicity Assay 

Approximately 10,000 VERO cells were cultured in 96-well plates and incubated with 150 μM of kaempferol or 150 μM MTZ for 40, 60, 90, and 180 min; VERO cells in the presence of DMSO and without kaempferol were used as controls. The total volume of each well was adjusted to 200 μL. Briefly, 95 μL of PBS (pH 6.8) + 5 μL of WST-1 (Sigma-Aldrich, MO, USA) was added to 100 μL of WST-1 in each well. The plate was then incubated and protected from light for 30 min at 37 °C. Then, 50 μL of the supernatant was taken from each well, placed in another 96-well plate, and finally read in a microplate reader Synergy HT (Biotek, VER, USA) at a wavelength of 450 nm.

This experiment was repeated three times and the data presented here are an average of the standard values. The percentage viability of VERO cells was determined by inserting the optical densities into the formula: %Viability = [(mean Optical Density (O.D) treated cells × 100]/(mean O.D. control cells) [44].

### 4.5. Real-Time qPCR of Antioxidant Enzymes of E. histolytica 

For the extraction of RNA from 15,000 *E. histolytica* trophozoites incubated with 150 μM of kaempferol or MTZ for 90 min at 37 °C, the TRIzol-Chloroform (Thermo Scientific, Waltham, MA, USA) method was used. At the end of this process, the cells were resuspended in RNAse-free water. RNA was stored at −80 °C until use. RNA quantification was performed using a Nanodrop Lite (Thermo Scientific, Waltham, MA, USA). The isolated RNA was treated with RQ1 RNase-Free DNase (PROMEGA, Fitchburg, WI, USA) to avoid genomic DNA contamination and cDNA was synthesized using the First Strand cDNA synthesis kit (Thermo Scientific, MA, USA) according to the manufacturer’s protocol. RT-qPCR was performed with a Step One Real-Time PCR system (Applied Biosystems, Foster City, CA, USA) by monitoring the increase in fluorescence in real time using SYBR Green PCR Master Mix (Applied Biosystems, CA, USA). Melting curve protocols were performed to ensure the specificity of the amplification products. The primers were designed using Primer Express 3.0.1 software (Applied Biosystems, Foster City, CA, USA) and synthesized commercially (IDT Integrated DNA Technologies, Coralville, IA, USA). The size of all amplicons was designed of 150 pb. (Table 1). For *E. histolytica* trophozoites, primers specific for Rr, Prx, TrxR, and Trx were used; as a control, Glyceraldehyde-3-phosphate dehydrogenase (GAPDH) from *E. histolytica* was used.

The application of the comparative Cycle Threshold (CT) method was conducted to validate the effect of treatment on the expression of two endogenous controls 18S subunit ribosomal and glyceraldehyde-3-phosphate dehydrogenase (GAPDH). We selected GAPDH because no statistically significant relationship was found between the treatment and basal expression of this gene. The relative quantification of antioxidant enzymes was calculated using the CT method by applying the comparative cycle threshold CT, which uses the arithmetic formula 2^−ΔΔCT^ [45]. To validate the method, we verified that the amplification efficiency for the target genes and the endogenous gene GAPDH were nearly equal, examined CT variations with serial of cDNA template dilutions, and a plot of log cDNA concentrations versus CT was made and efficiency was calculated using the equation E = −1 + 10^(−1/slope)^. The statistical significance between the untreated and treated trophozoites was calculated using Bonferroni‘s test with GraphPad Prism statistical software (GraphPad, San Diego, CA, USA).

### 4.6. Protein Extraction

Protein extraction was performed on 15,000 *E. histolytica* trophozoites grown in TYI-S-33 medium, incubated with 150 μM of kaempferol or 150 MTZ for 90 min at 37 °C, and then lysed in lysis buffer with complete proteases inhibitor cocktail (Roche, Basel, Switzerland); 100 mM PHMB (Sigma-Aldrich, MO, USA); 1 mM E64 (Thermo Scientific, MA, USA); 100 mM Tris (Sigma-Aldrich, MO, USA); 100 mM PMSF (Thermo Scientific, MA, USA); and 0.5 M iodoacetamide (Sigma-Aldrich, MO, USA). Three cycles of freezing and thawing were subsequently carried out, the supernatant was retrieved and centrifuged at 15,000× *g* for 5 min at 4 °C, the protein concentration was determined using Nanodrop Lite (Thermo scientific, MA, USA), and the protein integrity was determined by 10% SDS-PAGE and staining with Coomassie Blue (Sigma-Aldrich, MO, USA) [25].

### 4.7. Neutrophil Purification

For the isolation of neutrophils, we used male hamsters *(Mesocricetus auratus)*, an amoebiasis-susceptible model weighing approximately 100 g of six to eight weeks of age. Neutrophils were extracted from an enriched fraction, which was obtained from hamsters via cardiac puncture, using a 3 mL insulin syringe with heparin (Pisa, MEXICO, Ciudad de México, México). The heparinized blood was placed in a 15 mL conical tube. Subsequently, 6 mL of PolymorphoPrep (Sigma-Aldrich, MO, USA) was placed in another 15 mL tube, and 6 mL of blood was added into 15 mL tube. The tube was centrifuged at 758× *g* for 35 min at 20 °C. The supernatant was discarded, the pellet containing the neutrophil enriched fraction was obtained, and the neutrophil layer was collected. In both treatments, the remaining red blood cells were subjected to hypotonic shock with sterile 0.2% NaCl for 30 s and 1.6% NaCl for 1 min. Finally, neutrophils were resuspended in RPMI + SFB 10% (Sigma, MO, USA) or DMEM/F-12/Media + SFB 10%, incubated at 37 °C, and counted with the Türk staining method; a representative image of inactivated neutrophils is shown in Appendix A [46]. The number of neutrophils was counted in a Neubauer chamber, and their viability (viable cells excluding trypan blue) and purity (morphology) ascertained. Briefly, ≥95% of the purified cells were neutrophils, and the cells were adjusted to 300,000 neutrophils/mL [32]. The animals were handled in accordance with Mexican Federal Regulations for Animal Experimentation and Care (NOM-062-ZOO-1999, Ministry of Agriculture, México City, México). The institutional Animal Care and Use Committee from Escuela Superior de Medicina acted as the regulatory office for approving research protocols and approved the animal care and handling of the hamsters (protocol number ESM-CICUAL 07/26-08-2019). 

### 4.8. Neutrophils and E. histolytica Trophozoite Interactions

A total of 300,000 neutrophils were incubated with 15,000 *E. histolytica* trophozoites for 90 min at 37 °C in RPMI + SFB 10% or DMEM/F-12/Media + FBS 10% medium in a 24-well plate in the presence or absence of 150 μM of kaempferol or MTZ for 90 min at 37 °C. Neutrophils in the presence of LPS (20 μg/mL) (Sigma-Aldrich, MO, USA) and interactions in the presence of ABAH, an inhibitor of MPO at 100 μM/L (Sigma, St. Louis, MO, USA) and DMSO, were used. LPS was a positive control of the production of ROS and NO. On the other hand, we used ABAH as negative control, as it inhibited MPO activity. 

### 4.9. MPO Activity

MPO quantification was performed on the supernatants of the interactions for 20, 40, 60, and 90 min in RPMI + SFB 10% medium, from which 50 μL was transferred to 96-well plates, and 50 μL of 3,3′,5,5′-tetramethylbenzidine (TMB) (Sigma-Aldrich, MO, USA) was subsequently added followed immediately by 50 μL of H_2_O_2_ (5 mM/L in water, high purity) (Sigma-Aldrich, MO, USA). The mixture was allowed to react for 10 min until a color change was visualized. After, the reaction was stopped with 50 μL of 2.5 M/L sulfuric acid (Sigma-Aldrich, MO, USA). The plates were centrifuged at 600× *g* for 10 min, 200 μL of the supernatant from each well was transferred to another plate, and the optical density (OD) in each well was determined at 405 nm using a microplate reader. PMA was used as a positive control of MPO activity [32].

### 4.10. ROS and NO Production

ROS and NO production were determined using a ROS kit (Abcam, MA, USA) [47] and a NO assay kit (Abcam, MA, USA) [48]. The production of ROS and NO was quantified from the interaction of 300,000 neutrophils with 15,000 *E. histolytica* trophozoites with 150 μM of kaempferol or MTZ for 90 min at 37 °C in DMEM/F12 (Sigma-Aldrich, MO, USA) + 5% SFB (Sigma-Aldrich, MO, USA) for 90 min. LPS (20 μg/mL) was used as a positive control, and trophozoites in the absence of kaempferol and trophozoites with 0.05% DMSO were used as controls. At the end of the interactions, the supernatant was removed. Samples were previously deproteinized for the nitric oxide assay. For ROS and NO determination, the manufacturer’s specifications were followed. The plates were read in a microplate reader; ROS was read at a wavelength of 480 nm excitation/530 nm and the production of NO was read at a wavelength of 540 nm.

### 4.11. Statistical Analysis

All the data were processed using GraphPad Prism 8.0 software. Statistical analyses were performed using one-way ANOVA, two-way ANOVA, and Student’s *t*-test. If a significant (*p* < 0.05) main effect or association was identified, the respective group means were compared using the Bonferroni test.

## 5. Conclusions

In conclusion, in this study we confirmed that kaempferol is an effective drug against *E. histolytica* compared with MTZ at the same concentration, because kaempferol decreased its viability. In addition, we observed an overexpression of some *E. histolytica* proteins in the presence of kaempferol and a downregulation in the expression of Pr, Rr, and TrxR genes, which are antioxidant enzymes of the parasite. Interestingly, kaempferol acts both on *E. histolytica* trophozoites and on hamster neutrophils, regulating the production of ROS and ON as well as the activity of neutrophil MPO.

Further studies are necessary in clinical assays because kaempferol could be effective for patients with amoebiasis, as this flavonoid does not present adverse effects like MTZ does.

## Figures and Tables

**Figure 1 ijms-24-11216-f001:**
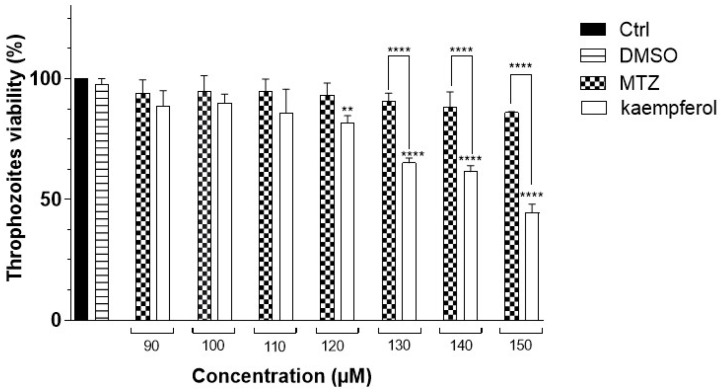
Trophozoite viability of *E. histolytica* at different times and concentrations of kaempferol or MTZ. Percentage of viability of trophozoites of *E. histolytica* in the presence of different concentrations of kaempferol or MTZ (90, 100, 110, 120, 130, 140, and 150 μM) for 90 min was determined with WST-1 assay. DMSO (Dimethylsulfoxide) was used as a vehicle. Data represent the mean ± SD (triplicate). *p*-values were determined with ANOVA (**** *p* < 0.0001; ** *p* < 0.01), compared with the control group (Ctrl), while those compared with the MTZ *p*-values were determined with ANOVA (**** *p* < 0.0001).

**Figure 2 ijms-24-11216-f002:**
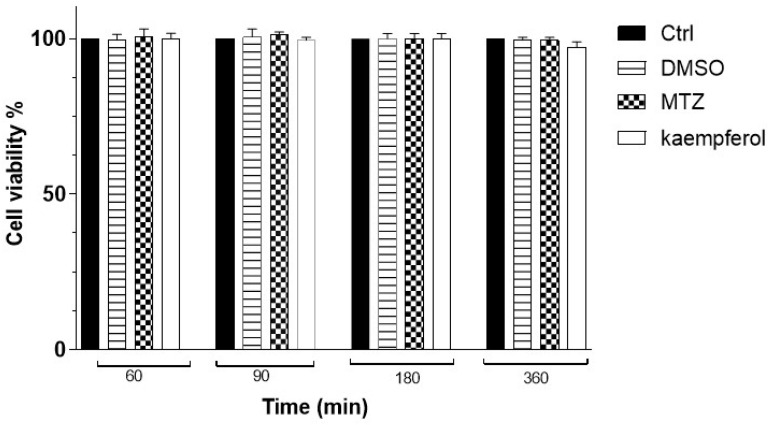
Percentage of VERO cells’ viability with kaempferol at 60, 90, 180, and 360 min was determined using WST-1 assay. Data represent the mean ± SD (triplicate). *p*-values were determined with ANOVA.

**Figure 3 ijms-24-11216-f003:**
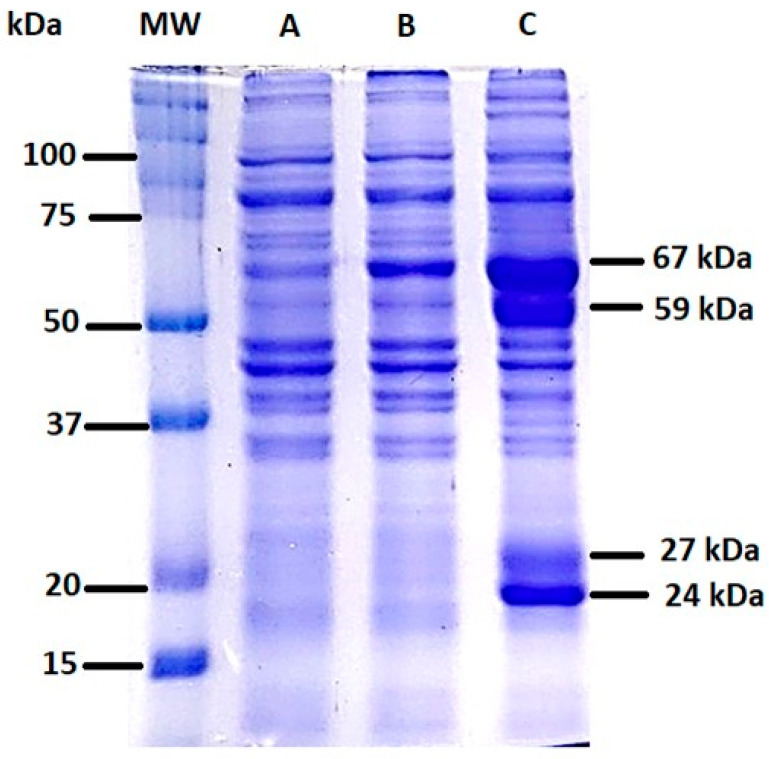
Cell lysate of *E. histolytica* incubated with kaempferol or MTZ at 150 μM for 90 min at 37 °C and electrophoresed on a 10% SDS-polyacrylamide gel. (A) Total cell lysate, (B) total cell lysate incubated with MTZ, (C) total cell lysate incubated with kaempferol for 90 min. MW = molecular weight.

**Figure 4 ijms-24-11216-f004:**
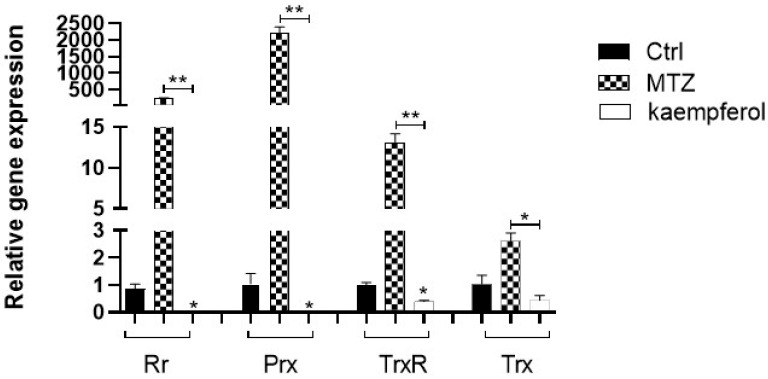
Differential expression of the antioxidant enzyme genes in trophozoites of *E. histolytica*. Gene expression of antioxidant enzymes (Rr, Prx, TrxR, and Trx) in trophozoites in the presence of kaempferol at 150 μM for 90 min was determined using qRT-PCR. There is a statistically significant difference between Rr, Prx, and TrxR genes in kaempferol compared with those in the control and the MTZ groups. Statistical analyses were determined with Student’s *t*-test (* *p* < 0.05; ** *p* < 0.01). Data represent the mean ± SD (triplicate). *p*-values were determined using Student’s *t*-test and using Bonferroni’s test.

**Figure 5 ijms-24-11216-f005:**
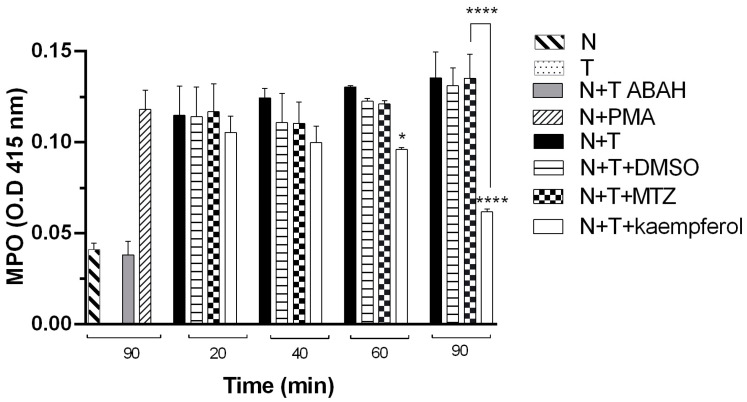
Determination of MPO activity in the interactions of *E. histolytica* and neutrophil of hamsters. MPO activity in neutrophils in the presence of trophozoites of *E. histolytica* incubated with kaempferol or MTZ at 150 μM at different times was determined. DMSO was used as a vehicle, PMA (phorbol 12-myristate 13-acetate) was used as a positive control, and MPO inhibitor 100 mM (ABAH) was used as MPO activity control. Data represent the mean ± SD of the three independent experiments. *p*-values were determined using two-way ANOVA (**** *p* < 0.0001; * *p* < 0.05), comparing N+T with the N+T+kaempferol group, while compared with the MTZ group, *p*-values were determined using ANOVA (**** *p* < 0.0001).

**Figure 6 ijms-24-11216-f006:**
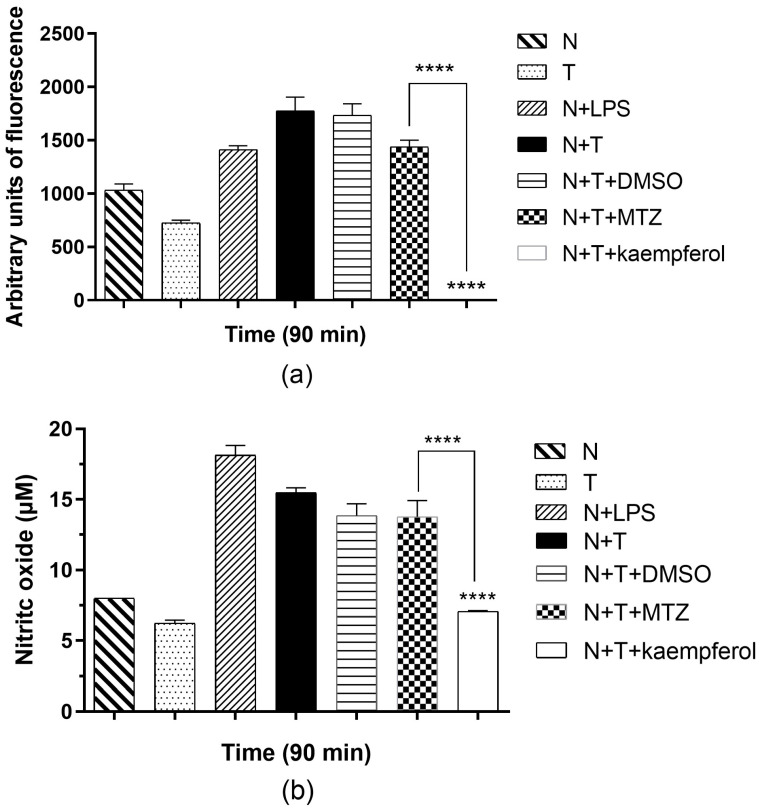
Determination of ROS and NO produced in the interaction between neutrophils and trophozoites of *E. histolytica* in the presence of kaempferol. (**a**) Quantification of total ROS using dichlorodihydrofluorescein diacetate (DCFH-DA) assay kit in the interaction of neutrophils in presence of trophozoites incubated with kaempferol at 150 μM for 90 min. DMSO and LPS were used as controls. *p*-values were determined using two-way ANOVA (**** *p* < 0.0001) to compare the N+T group with the N+T+kaempferol group, while compared with the MTZ group, *p*-values were determined using two-way ANOVA (**** *p* < 0.0001). (**b**) Detection of NO using the Griess method in the interaction of neutrophils in the presence of *E. histolytica* incubated with kaempferol at 150 μM for 90 min. Data represent the mean ± SD of the three independent experiments. *p*-values were determined using two-way ANOVA (**** *p* < 0.0001), comparing the N+T group with the N+T+kaempferol group, while compared with the MTZ group, *p*-values were determined using ANOVA (**** *p* < 0.0001).

**Table 1 ijms-24-11216-t001:** List of primer sequences and PCR conditions for RT-PCR.

Gene	Accession	Forward (5′--3′)	Reverse (5′--3′)
Thioredoxin (Trx)	XM_649815.1	TATGCAGAGTGGTG- TGGTCCAT	AAATGTCGGCATAC-AACGAATACC
Thioredoxin reductase (TrxR)	XM_650656.2	ATGAGAACACAATC- AGAGAAGTATGGA	AGCTGTAGCACCTG-TTGCAATAAT
Peroxiredoxin (Prx)	XM_646911.2 XM_644418.2 XM_643430.2	CGAAGCAGGAATTG- CAAGAAG	GCTCCATGTTCATC- ACTGAATTG
Rubrerythrin (Rr)	XM_647039.2	CATGCTCAAATTGC- TGCTAGACTT	ATATCCACATTCTC- TACAAACCCAAA
GAPDH	AB002800.1	TTCATGGATCCAAA- ATACATGGTT	GCCAATTTGAGCTG-GATCTCTT

## Data Availability

The data are contained within the article.

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
