# Peer review of "In Vitro Evaluation of the Antiamoebic Activity of Kaempferol against Trophozoites of Entamoeba histolytica and in the Interactions of Amoebae with Hamster Neutrophils"

_ijms, 2023, doi:10.3390/ijms241311216_

Round 1

Reviewer 1 Report

The reviewer believes that this manuscript needs a very thorough revision. The reviewer is not entirely sure, but this manuscript needs English editing because some of the authors' thoughts are not clear. For example, in lines 46–50 the authors write: “E. histolytica in the liver can induce the formation of lesions in the hepatic parenchyma presenting an important inflammatory response, which is constituted by polymorphonuclear leukocytes (PMNs), mainly neutrophils (acute phase), as well as by mononuclear cells such as macrophages (chronic phase)". Here, apparently, we are talking about the fact that E. histolytica is the cause of inflammation? Which is accompanied by an increase in the number of immune cells of one type or another, depending on the course of the infection? Or should we be talking about the fact that E. histolytica can spread through the circulation in the portal vein system and cause necrotic liver abscesses, etc.?

1. Abstract.

In lines 22-23, the authors write "...amoebic viability of trophozoites..."; is it correct to say so? Maybe the authors had in mind the survival of trophozoites?

Lines 27-29; here we should probably talk about the violation of the regulation of the synthesis of antioxidant enzymes in E. histolytica? So there is in the "results" section, which shows a decrease in the expression of the genes of these enzymes.

2. Section "Introduction"

The reviewer believes that this section should be improved. At present, the purpose of the study does not become apparent from this section. Perhaps data should be drawn here that would enable the reader to understand how kaempferol affects not only E. histolytica, but also immune cells.

3. The section "materials and methods" requires a significant revision. Many methods are described in such a way that it will be difficult for other researchers to understand and reproduce them. Interspersed are descriptions of the cell cultures used and methods of working with them. This way of presentation makes it difficult to understand this section. Many of the subsections in this section are missing references to the methods used. In many subsections, the number of cells used for experiments is indicated, but it is not described by what method their number was determined.

4. Subsection 4.1, lines 316-317, where the authors write "The trophozoites were harvested at the end of the logarithmic growth phase (48 hours) by chilling to 4 °C". From the protocols known to the reviewer, it follows that cooling is used to detach the trophozoites from the walls of the test tube, and then centrifugation. Can you write like that?

5. Subsection 4.2, lines 326-333, this subsection is very far in the text from subsection 4.8, which describes a method for isolating and purifying neutrophils. At this point, there is information here only that the animal study was performed in accordance with the requirements of the regulatory authorities. The title of the subsection itself is incorrect; the authors call this section "Animals of experimentation", probably, it should be about experimental animals.

6. Subsection 4.3, lines 335-340 is like a short review of the literature. The section itself refers to the assessment of the viability of only E. histolytica, but the title of the subsection implies the assessment of the viability of any cells.

7. Subsection 4.4, lines 355-360; it is unclear why the authors assessed the viability of E. histolytica using trypan blue. Prior to this, the formazan reduction method was used (see subsection 4.3). Why use two different methods to measure the same property? Moreover, the results section (Figure 1) describes the effect of kaempferol and metronidazole on the viability of E. histolytica as a function of time and concentration. Obviously, in such cases, one method should be used, or explain why two different ones should be used.

8. Subsection 4.9, lines 424-427, here it is necessary to somehow redo this sentence, it is not clear what this is about. It seems here that neutrophils treated with lipopolysaccharide, or a myeloperoxidase inhibitor were used as controls. More: ABAH is 4-Aminobenzoic acid hydrazide? Not 4-amino-benzoic acid hydrobenzoic acid?

9. Section "Results"

10. Subsection 2.1. The subsection title is too long. It could be called "Effect of kaempferol on the survival of E. histolytica and VERO cells".

Figure 1b shows that the survival rate of E. histolytica is reduced to 44.5%, then there is a decrease of 55.5% (where 90 minutes); the authors write “…we observed a decrease of 44.5% in trophozoite viability at 90 min, …” (lines 127-128). The authors should be more careful about the digital data in the description of the figures.

Also in this subsection, the authors show surprisingly high concentrations of metronidazole, which are necessary to reduce the viability of E. histolytica; other researchers indicated lower concentrations and separately noted that in order for E. histolytica isolates to be resistant to metronidazole (~20-40 μM), they must be cultivated for a sufficiently long time in increasing concentrations of the drug (see, for example, Wassmann, C. , Hellberg, A., Tannich, E., & Bruchhaus, I. (1999) Metronidazole resistance in the protozoan parasite Entamoeba histolytica is associated with increased expression of iron-containing superoxide dismutase and peroxiredoxin and decreased expression of ferredoxin 1 and flavin reductase Iyer LR, Singh N, Verma AK, Paul J. Differential expression and immunolocalization of antioxidant enzymes in Entamoeba histolytica isolates during metronidazole stress Biomed Res Int. 2014;2014 :704937. doi: 10.1155/2014/704937.).

11. Subsection 2.3. The data in Figure 4 in this subsection would be better presented in the form of a table with the obtained values. Due to the wide range, it is difficult to perceive small values of gene expression.

12. Subsection 2.4. Figure 5 shows PMA, which was used as a positive control. There is nothing about this in the "materials and methods" section, only lipopolysaccharide is indicated there. Does the reviewer correctly understand that kaempferol reduces myeloperoxidase activity? If so, then this substance should increase the survival of E. histolytica when they interact with neutrophils.

13. Section "Discussion".

While reading the "results" section, the reviewer was struck by the high concentrations of metronidazole that were used in the experiments and had no significant effect on E. histolytica. There seems to be some kind of error in this, and this is the reason for the erroneous conclusions.

The authors carefully avoid discussing the implications of kaempferol's effect on neutrophils; according to their results, kaempferol inhibits both myeloperoxidase activity and the formation of reactive oxygen species and NO. That is, kaempferol reduces the effectiveness of neutrophils in the fight against E. histolytica? Metronidazole has no such effects on neutrophils. These facts seem to argue against kaempferol's high potential for use against E. histolytica in vivo because the first line of immune defense is blocked.

At the same time, for example, antioxidant enzymes, in particular E. histolytica peroxiredoxin, are known to play a role in the survival of trophozoites in the presence of ROS (see, for example, Sen, A., Chatterjee, N. S., Akbar, M. A., Nandi, N. , & Das, P. (2007) The 29-kilodalton thiol-dependent peroxidase of Entamoeba histolytica is a factor involved in pathogenesis and survival of the parasite during oxidative stress Eukaryotic Cell, 6(4), 664-673.).

It is possible that reducing the expression of antioxidant enzymes with kaempferol will increase the sensitivity of E. histolytica to reactive oxygen species, but this will not increase the antiprotozoal activity of neutrophils, because they are also inactivated.

In general, it's difficult.

Author Response

Reviewer 1.

The reviewer believes that this manuscript needs a very thorough revision. The reviewer is not entirely sure, but this manuscript needs English editing because some of the authors' thoughts are not clear. For example, in lines 46–50 the authors write: “E. histolytica in the liver can induce the formation of lesions in the hepatic parenchyma presenting an important inflammatory response, which is constituted by polymorphonuclear leukocytes (PMNs), mainly neutrophils (acute phase), as well as by mononuclear cells such as macrophages (chronic phase)". Here, apparently, we are talking about the fact that E. histolytica is the cause of inflammation? Which is accompanied by an increase in the number of immune cells of one type or another, depending on the course of the infection? Or should we be talking about the fact that E. histolytica can spread through the circulation in the portal vein system and cause necrotic liver abscesses, etc.?

R= We are very grateful to you for the careful reading of our manuscript. Since many years ago it is known that E. histolytica trophozoites causes an acute and chronic inflammation with the presence of different types of inflammatory cells as polymorphonuclear cells in acute inflammatory stage as well as mononuclear cells in chronic stage, it has been demonstrated by numerous studies that the inflammation is a relevant mechanism of liver damage (Olivos-García et al. 2007; Tsutsumi and Shibayama 2006; Olivos-García et al. 2004, Tsutsumi and Martinez-Palomo 1988; Tsutsumi et al. 1984).

Olivos-García, A. et al. 2007. “Late Experimental Amebic Liver Abscess in Hamster Is Inhibited by Cyclosporine and N-Acetylcysteine.” Experimental and Molecular Pathology 82(3): 310–15.

Tsutsumi, Víctor, and Mineko Shibayama. 2006. “Experimental Amebiasis: A Selected Review of Some in Vivo Models.” Archives of Medical Research 37(2): 210–20.

Olivos-García, A. et al. 2004. “Inflammation, Complement, Ischemia and Amoebic Survival in Acute Experimental Amoebic Liver Abscesses in Hamsters.” Experimental and Molecular Pathology 77(1): 66–71.

Tsutsumi, V., and A. Martinez-Palomo. 1988. “Inflammatory Reaction in Experimental Hepatic Amebiasis. An Ultrastructural Study.” American Journal of Pathology 130(1): 112–19.

Tsutsumi, V., R. Mena-Lopez, F. Anaya-Velazquez, and A. Martinez-Palomo. 1984. “Cellular Bases of Experimental Amebic Liver Abscess Formation.” American Journal of Pathology 117(1): 81–91.

In lines 22-23, the authors write "...amoebic viability of trophozoites..."; is it correct to say so? Maybe the authors had in mind the survival of trophozoites?

R= Thank you for your comment. Numerous manuscripts on the area report the survival of amoeba as amoebic viability (Herrera-Martínez et al. 2022; Rangel-Castañeda et al. 2018; Pais-Morales et al. 2016; Carrero et al. 2006).

Herrera-Martínez, Mayra et al. 2022. “Alpha-Terthienyl Increases Filamentous Actin of Entamoeba Histolytica.” Molecular and Biochemical Parasitology 252: 111512.

Rangel-Castañeda, Itzia Azucena et al. 2018. “Amoebicidal Activity of Curcumin on Entamoeba Histolytica Trophozoites.” Journal of Pharmacy and Pharmacology 70(3): 426–33.

Pais-Morales, Jonnatan et al. 2016. “Resveratrol Induces Apoptosis-like Death and Prevents in Vitro and in Vivo Virulence of Entamoeba Histolytica.” PLoS ONE 11(1): 1–23.

Carrero, Julio C. et al. 2006. “Dehydroepiandrosterone Decreases While Cortisol Increases in Vitro Growth and Viability of Entamoeba Histolytica.” Microbes and Infection 8(2): 323–31. 

Lines 27-29; here we should probably talk about the violation of the regulation of the synthesis of antioxidant enzymes in E. histolytica? So, there is in the "results" section, which shows a decrease in the expression of the genes of these enzymes.

R= Thank you for your comment. You are right we have changed the sentence at 27-28 lines. “In conclusion, we confirmed that kaempferol is an effective drug against E. histolytica through the decrease of E. histolytica antioxidant enzymes expression and a regulator of several neutrophil mechanisms, such as MPO activity and the regulation of ROS and NO”.

  1. Section "Introduction"

The reviewer believes that this section should be improved. At present, the purpose of the study does not become apparent from this section. Perhaps data should be drawn here that would enable the reader to understand how kaempferol affects not only E. histolytica, but also immune cells.

R= You are right, as you suggested we added a brief comment in the introduction section regarding the kaempferol effects on immune cells (Lines 100-103). This is a topic addressed by different authors (Zeng et al. 2020; Cao et al. 2014; Chen et al. 2012). “Kaempferol has been reported to have an effect in suppressing ROS production in mouse bone marrow-derived neutrophils [24], moreover in a model of induced mouse mastitis [25], and a model of mouse model of LPS-induced acute lung injury treats with kaempferol, the MPO activation was reduced[26]”.

Zeng, Jie et al. 2020. “Kaempferol Blocks Neutrophil Extracellular Traps Formation and Reduces Tumour Metastasis by Inhibiting ROS-PAD4 Pathway.” Journal of Cellular and Molecular Medicine 24(13): 7590–99.

Cao, Rongfeng et al. 2014. “Protective Effects of Kaempferol on Lipopolysaccharide-Induced Mastitis in Mice.” Inflammation 37(5): 1453–58.

Chen, Xiaojun et al. 2012. “Kaempferol Regulates MAPKs and NF-ΚB Signaling Pathways to Attenuate LPS-Induced Acute Lung Injury in Mice.” International Immunopharmacology 14(2): 209–16.

  1. The section "materials and methods" requires a significant revision. Many methods are described in such a way that it will be difficult for other researchers to understand and reproduce them. Interspersed are descriptions of the cell cultures used and methods of working with them. This way of presentation makes it difficult to understand this section. Many of the subsections in this section are missing references to the methods used. In many subsections, the number of cells used for experiments is indicated, but it is not described by what method their number was determined.

R= Thank you for your comment. As you requested in material and method, now we have added more detailed information to clarify the methodology. We have added new references in the methods sections where they are missing. Additionally, the method used to determine the number of cells was also added (Line 552).

Amoebic cultures reference: Diamond, L.S.; Harlow, D.R.; Cunnick, C.C. A New Medium for the Axenic Cultivation of Entamoeba histolytica and Other Entamoeba. Trans. R. Soc. Trop. Med. Hyg. 1978, 72, 431–432, doi:10.1016/0035-9203(78)90144-X.

VERO cell culture reference: Jabari, S.; Keshavarz, H.; Salimi, M.; Morovati, H.; Mohebali, M.; Shojaee, S. In Vitro Culture of Toxoplasma Gondii in Hela, Vero, RBK and A549 Cell Lines. Infez. Med. 2018, 26, 145–147.

  1. histolytica viability assays (WST-1) assay references:

Redondo, M.J.; Palenzuela, O.; Alvarez-Pellitero, P. In Vitro Studies on Viability and Proliferation of Enteromyxum Scophthalmi (Myxozoa), an Enteric Parasite of Cultured Turbot Scophthalmus Maximus. Dis. Aquat. Organ. 2003, 55, 133–144

Soares, C.O.; Colli, W.; Bechara, E.J.H.; Alves, M.J.M. 1,4-Diamino-2-Butanone, a Putrescine Analogue, Promotes Redox Imbalance in Trypanosoma Cruzi and Mammalian Cells. Arch. Biochem. Biophys. 2012, 528, 103–110

VERO cell WST-1 cytotoxicity assay reference: Muiva-Mutisya, L.M.; Atilaw, Y.; Heydenreich, M.; Koch, A.; Akala, H.M.; Cheruiyot, A.C.; Brown, M.L.; Irungu, B.; Okalebo, F.A.; Derese, S.; et al. Antiplasmodial Prenylated Flavanonols from Tephrosia Subtriflora. Nat. Prod. Res. 2018, 32, 1407–1414

Real-time qPCR of antioxidant enzymes of E. histolytica reference: Livak, K.J.; Schmittgen, T.D. Analysis of Relative Gene Expression Data Using Real-Time Quantitative PCR and the 2-ΔΔCT Method. Methods 2001, 25, 402–408

ROS and NO production references: Xiao, L.; Ma, X.; Ye, L.; Su, P.; Xiong, W.; Bi, E.; Wang, Q.; Xian, M.; Yang, M.; Qian, J.; et al. IL-9/STAT3/Fatty Acid Oxidation-Mediated Lipid Peroxidation Contributes to Tc9 Cell Longevity and Enhanced Antitumor Activity. J. Clin. Invest. 2022, 132, 1–16, doi:10.1172/JCI153247.

El-Mancy, E.M.; Elsherbini, D.M.A.; Al-Serwi, R.H.; El-Sherbiny, M.; Ahmed Shaker, G.; Abdel-Moneim, A.M.H.; Enan, E.T.; Elsherbiny, N.M. α-Lipoic Acid Protects against Cyclosporine A-Induced Hepatic Toxicity in Rats: Effect on Oxidative Stress, Inflammation, and Apoptosis. Toxics 2022, 10, doi:10.3390/toxics10080442.

  1. Subsection 4.1, lines 316-317, where the authors write "The trophozoites were harvested at the end of the logarithmic growth phase (48 hours) by chilling to 4 °C". From the protocols known to the reviewer, it follows that cooling is used to detach the trophozoites from the walls of the test tube, and then centrifugation. Can you write like that?

R = Thank you for your observation. The amoeba adheres to plastic and glass, so to detach them it is necessary to cool them to 4oC. It is a method commonly used by researchers in the area. This methodology is well known since many years ago and it has been reported by many researchers (Pulido-Ortega et al. 2019; Carrero et al. 2006; Choi et al. 2005; Bruchhaus, Richter, and Tannich 1997; Leroy et al. 1995). We write as request in methodology (Line 384) “The trophozoites were cooled, detached and harvested at the end of the logarithmic growth phase (48 hours) by chilling to 4 °C”.

Pulido-Ortega, J. et al. 2019. “Functional Characterization of an Interferon Gamma Receptor-Like Protein on Entamoeba Histolytica Julieta.” (July): 1–15.

Carrero, Julio C. et al. 2006. “Dehydroepiandrosterone Decreases While Cortisol Increases in Vitro Growth and Viability of Entamoeba Histolytica.” Microbes and Infection 8(2): 323–31.

Choi, Min Ho et al. 2005. “An Unusual Surface Peroxiredoxin Protects Invasive Entamoeba Histolytica from Oxidant Attack.” Molecular and Biochemical Parasitology 143(1): 80–89.

Bruchhaus, Iris, Symi Richter, and Egbert Tannich. 1997. “Removal of Hydrogen Peroxide by the 29 KDa Protein of Entamoeba Histolytica.” Biochemical Journal 326(3): 785–89.

Leroy, A. et al. 1995. “Contact-Dependent Transfer of the Galactose-Specific Lectin of Entamoeba Histolytica to the Lateral Surface of Enterocytes in Culture.” Infection and Immunity 63(11): 4253–60.

  1. Subsection 4.2, lines 326-333, this subsection is very far in the text from subsection 4.8, which describes a method for isolating and purifying neutrophils. At this point, there is information here only that the animal study was performed in accordance with the requirements of the regulatory authorities. The title of the subsection itself is incorrect; the authors call this section "Animals of experimentation", probably, it should be about experimental animals.

R= We thank the reviewer for this observation. We have combined the subsection 4.2 and 4.8. We expect the section to improve. Lines 540-541 and lines 553-560.

  1. Subsection 4.3, lines 335-340 is like a short review of the literature. The section itself refers to the assessment of the viability of only E. histolytica, but the title of the subsection implies the assessment of the viability of any cells.

R= We thank the reviewer for this observation. We have modified the subsection the short review has been deleted and title were changed.

  1. Subsection 4.4, lines 355-360; it is unclear why the authors assessed the viability of E. histolytica using trypan blue. Prior to this, the formazan reduction method was used (see subsection 4.3). Why use two different methods to measure the same property? Moreover, the results section (Figure 1) describes the effect of kaempferol and metronidazole on the viability of E. histolytica as a function of time and concentration. Obviously, in such cases, one method should be used, or explain why two different ones should be used.

R= The method commonly used to determine the viability of amoebae is trypan blue (Rangel-Castañeda et al. 2022; Rangel-Castañeda et al. 2018; Tan et al. 2010), however, several studies currently determine amoeba viability through the use of tetrazolium salt (Herrera-Martínez et al. 2022; Díaz-Godínez et al. 2019; Mukhopadhyay and Chaudhuri 1996). We wanted to show that the results of both methods are the same. Now it has been eliminated the trypan blue results.

Rangel-Castañeda, Itzia Azucena et al.  2022. “Drug Repositioning: Antiprotozoal Activity of Terfenadine against Entamoeba Histolytica Trophozoites.” Parasitology Research 121(1): 303–9.

Rangel-Castañeda, Itzia Azucena et al. 2018. “Amoebicidal Activity of Curcumin on Entamoeba Histolytica Trophozoites.” Journal of Pharmacy and Pharmacology 70(3): 426–33.

Tan, Z. N. et al. 2010. “Identification of Entamoeba Histolytica Trophozoites in Fresh Stool Sample: Comparison of Three Staining Techniques and Study on the Viability Period of the Trophozoites.” Tropical Biomedicine 27(1): 79–88.

Herrera-Martínez, Mayra et al. 2022. “Alpha-Terthienyl Increases Filamentous Actin of Entamoeba Histolytica.” Molecular and Biochemical Parasitology 252: 111512.

Díaz-Godínez, César et al. 2019. “Synthetic Bovine Lactoferrin Peptide Lfampin Kills Entamoeba Histolytica Trophozoites by Necrosis and Resolves Amoebic Intracecal Infection in Mice.” Bioscience Reports 39(1): 1–16.

Mukhopadhyay, R M, and S K Chaudhuri. 1996. “Rapid in Vitro Test for Determination of Anti-Amoebic Activity.” Transactions of the Royal Society of Tropical Medicine and Hygiene: 189–91.

  1. Subsection 4.9, lines 424-427, here it is necessary to somehow redo this sentence, it is not clear what this is about. It seems here that neutrophils treated with lipopolysaccharide, or a myeloperoxidase inhibitor were used as controls. More: ABAH is 4-Aminobenzoic acid hydrazide? Not 4-amino-benzoic acid hydrobenzoic acid?

R= We thank the reviewer for this comment, numerous studies have documented that LPS stimulates the activity of neutrophils (Pan et al. 2022; Li et al. 2016). Previous work by our group demonstrated that Entamoeba histolytica stimulates myeloperoxidase enzyme activity in neutrophils and that ABAH inhibits this activity (Contis Montes de Oca et al. 2020; Cruz-Baquero et al. 2017). By one hand, we used LPS as positive control of ROS and NO release and on the other hand, we used ABAH as negative control of MPO activity since Entamoeba histolytica induces MPO activity and in the presence of the inhibitor ABAH this activity is suppressed. We have added a brief description (Lines 575-577) “LPS is a positive control of the production of ROS and NO. On the other hand, we use ABAH as negative control, as it inhibited MPO activity”. The correct name of ABAH is 4-amino-benzoic acid hydrazide, this was corrected in the manuscript (Line 574) “and interactions in the presence of ABAH (4-amino-benzoic acid hydrazide), an”.

Pan, Tingting et al. 2022. “Immune Effects of PI3K/Akt/HIF-1α-Regulated Glycolysis in Polymorphonuclear Neutrophils during Sepsis.” Critical Care 26(1): 1–17.

Li, Yan et al. 2016. “B7H3 Ameliorates LPS-Induced Acute Lung Injury via Attenuation of Neutrophil Migration and Infiltration.” Scientific Reports 6(August): 1–10.

Contis Montes de Oca, Arturo et al. 2020. “Neutrophil Extracellular Traps and MPO in Models of Susceptibility and Resistance against Entamoeba Histolytica.” Parasite Immunology 42(6): 1–12.

Cruz-Baquero, Andrea et al. 2017. “Different Behavior of Myeloperoxidase in Two Rodent Amoebic Liver Abscess Models.” PLoS ONE 12(8): 1–23.

  1. Section "Results"
  2. Subsection 2.1. The subsection title is too long. It could be called "Effect of kaempferol on the survival of E. histolytica and VERO cells".

R= We thank the reviewer for the suggestion, now we have changed the subsection title.

  1. Figure 1b shows that the survival rate of E. histolytica is reduced to 44.5%, then there is a decrease of 55.5% (where 90 minutes); the authors write “…we observed a decrease of 44.5% in trophozoite viability at 90 min, …” (lines 127-128). The authors should be more careful about the digital data in the description of the figures.

R= You are right, we now corrected the sentence, and we also change by “the amebic viability is of 44.5 %” (Line 128) “viability of 44.5% with kaempferol at 150 µM. We also observed a significant difference in the viability of kaempferol compared with MTZ at these concentrations (Figure 1)”.

13.Also in this subsection, the authors show surprisingly high concentrations of metronidazole, which are necessary to reduce the viability of E. histolytica; other researchers indicated lower concentrations and separately noted that in order for E. histolytica isolates to be resistant to metronidazole (~20-40 μM), they must be cultivated for a sufficiently long time in increasing concentrations of the drug (see, for example, Wassmann, C. , Hellberg, A., Tannich, E., & Bruchhaus, I. (1999) Metronidazole resistance in the protozoan parasite Entamoeba histolytica is associated with increased expression of iron-containing superoxide dismutase and peroxiredoxin and decreased expression of ferredoxin 1 and flavin reductase Iyer LR, Singh N, Verma AK, Paul J. Differential expression and immunolocalization of antioxidant enzymes in Entamoeba histolytica isolates during metronidazole stress Biomed Res Int. 2014;2014 :704937. doi: 10.1155/2014/704937).

R= Thank for your comment. However, our experiments were done in a short time because as it is known amoeba kills neutrophils in vitro, even when the relation between amoeba neutrophils was 1:20. What you are referring to is about experiments that take a long time different to ours. The literature shows that metronidazole is used according to different parameters such as the number of trophozoites, time of incubation and the comparison among metronidazole with different drugs. In our case we use metronidazole with kaempferol at the same concentrations in order to compare them.

Reference

Model

Concentration MTZ

Times of experimentation

(Villegas-Gómez et al. 2021)

E. histolytica in vitro

0.25 mg/ml

24, 48 and 72 h

(Díaz-Godínez et al. 2020)

E. histolytica in vitro

6.8 µM

24 h

(Montaño et al. 2020)

E. histolytica in vitro

0.25 μg/ml

48 h

(Mohammed et al. 2019)

E. histolytica in vitro

0.27 μg/ml

24, 48 and 72 h

(Wassmann et al. 1999)

E. histolytica in vitro

40 µM

24 h

Villegas-Gómez, Isaac et al. 2021. “The Dichloromethane Fraction of Croton Sonorae, A Plant Used in Sonoran Traditional Medicine, Affect Entamoeba Histolytica Erythrophagocytosis and Gene Expression.” Frontiers in Cellular and Infection Microbiology 11(July): 1–9

Díaz-Godínez, César et al. 2020. “Anti-Amoebic Activity of Leaf Extracts and Aporphine Alkaloids Obtained from Annona Purpurea.” Planta Medica 86(6): 425–33.

Montaño, S. et al. 2020. “Vorinostat, a Possible Alternative to Metronidazole for the Treatment of Amebiasis Caused by Entamoeba Histolytica.” Journal of Biomolecular Structure and Dynamics 38(2): 597–603.

Mohammed, Seif Eldin A. et al. 2019. “In Vitro Activity of Some Natural Honeys against Entamoeba Histolytica and Giardia Lamblia Trophozoites.” Saudi Journal of Biological Sciences 26(2): 238–43.

Wassmann, Claudia, Andrea Hellberg, Egbert Tannich, and Iris Bruchhaus. 1999. “Metronidazole Resistance in the Protozoan Parasite Entamoeba Histolytica Is Associated with Increased Expression of Iron-Containing Superoxide Dismutase and Peroxiredoxin and Decreased Expression of Ferredoxin 1 and Flavin Reductase.” Journal of Biological Chemistry 274(37): 26051–56.

  1. Subsection 2.3. The data in Figure 4 in this subsection would be better presented in the form of a table with the obtained values. Due to the wide range, it is difficult to perceive small values of gene expression.

R= Thank you for your suggestion, however, we consider that Figure 4 clearly shows the differences in enzyme expression between the working groups.

  1. Subsection 2.4. Figure 5 shows PMA, which was used as a positive control. There is nothing about this in the "materials and methods" section, only lipopolysaccharide is indicated there. Does the reviewer correctly understand that kaempferol reduces myeloperoxidase activity? If so, then this substance should increase the survival of E. histolytica when they interact with neutrophils.

R= You are right, now we add the information about PMA in subsection 4.9 (Lines 587-588) “(OD) in each well was determined at 405 nm using a microplate reader. PMA (phorbol 12-myristate 13-acetate) was used as a positive control of MPO activity [34]”. On the other hand, you interpreted correctly about that kaempferol reduces MPO activity, however, kaempferol also at the same time acts on trophozoites of Entamoeba histolytica damaging them directly. Kaempferol has a dual activity.

  1. Section "Discussion"

While reading the "results" section, the reviewer was struck by the high concentrations of metronidazole that were used in the experiments and had no significant effect on E. histolytica. There seems to be some kind of error in this, and this is the reason for the erroneous conclusions.

R= As we explained in question 10; our experiments were done in a short time because as it is known amoeba kills neutrophils in vitro, even though the relation between amoeba neutrophils is 1:20. What you are referring to is about experiments that take a long time different to ours. The literature shows (see previous table) that metronidazole is used according to different parameters such as the number of trophozoites, time of incubation and the comparison among metronidazole with different drugs. In our case we use metronidazole with kaempferol at the same concentrations in order to compare them. Furthermore, at the times used, metronidazole did not kill E. histolytica trophozoites.

The authors carefully avoid discussing the implications of kaempferol's effect on neutrophils; according to their results, kaempferol inhibits both myeloperoxidase activity and the formation of reactive oxygen species and NO. That is, kaempferol reduces the effectiveness of neutrophils in the fight against E. histolytica? Metronidazole has no such effects on neutrophils. These facts seem to argue against kaempferol's high potential for use against E. histolytica in vivo because the first line of immune defense is blocked.

R= In the discussion section, we did not avoid talking about the effect of kaempferol on MPO, ROS, RNS and NO of neutrophils. In fact, we talked about several papers that refer to the effect of kaempferol on these enzymes and neutrophil products ( Sharma et al. 2021; Q. Li et al. 2018; Cao et al. 2014). Therefore, as we have described above, it has a dual effect on neutrophils and on amoeba. As was mentioned in the question 1 in the hamster susceptible model of amoebic liver abscess the inflammation caused by the amoeba induce liver damage (Olivos-García et al. 2007; Olivos-García et al. 2004; Tsutsumi and Shibayama 2006; Tsutsumi and Martinez-Palomo 1988; Tsutsumi et al. 1984). On the contrary, to what you mention, the activity of kaempferol in a susceptible model such as occurs in human infection could favor the resolution of liver damage since it has been demonstrated that this compound has an antioxidant activity and anti-inflammatory effect (Bian et al. 2022; Sharma et al. 2021; Zeng et al. 2020; Chen et al. 2012; Calderón-Montaño et al. 2011).

Sharma, Nidhi et al. 2021. “Antioxidant Role of Kaempferol in Prevention of Hepatocellular Carcinoma.” Antioxidants 10(9): 1–17.

Li, Qinchen et al. 2018. “Kaempferol Protects Ethanol-Induced Gastric Ulcers in Mice via pro-Inflammatory Cytokines and NO.” Acta Biochimica et Biophysica Sinica 50(3): 246–53.

Cao, Rongfeng et al. 2014. “Protective Effects of Kaempferol on Lipopolysaccharide-Induced Mastitis in Mice.” Inflammation 37(5): 1453–58.

Olivos-García, A. et al. 2007. “Late Experimental Amebic Liver Abscess in Hamster Is Inhibited by Cyclosporine and N-Acetylcysteine.” Experimental and Molecular Pathology 82(3): 310–15.

Olivos-García, A. et al. 2004. “Inflammation, Complement, Ischemia and Amoebic Survival in Acute Experimental Amoebic Liver Abscesses in Hamsters.” Experimental and Molecular Pathology 77(1): 66–71.

Tsutsumi, Víctor, and Mineko Shibayama. 2006. “Experimental Amebiasis: A Selected Review of Some in Vivo Models.” Archives of Medical Research 37(2): 210–20.

Tsutsumi, V., and A. Martinez-Palomo. 1988. “Inflammatory Reaction in Experimental Hepatic Amebiasis. An Ultrastructural Study.” American Journal of Pathology 130(1): 112–19.

Tsutsumi, V., R. Mena-Lopez, F. Anaya-Velazquez, and A. Martinez-Palomo. 1984. “Cellular Bases of Experimental Amebic Liver Abscess Formation.” American Journal of Pathology 117(1): 81–91.

Bian, Yifei et al. 2022. “Kaempferol Reduces Obesity, Prevents Intestinal Inflammation, and Modulates Gut Microbiota in High-Fat Diet Mice.” Journal of Nutritional Biochemistry 99(2): 108840.

Zeng, Jie et al. 2020. “Kaempferol Blocks Neutrophil Extracellular Traps Formation and Reduces Tumour Metastasis by Inhibiting ROS-PAD4 Pathway.” Journal of Cellular and Molecular Medicine 24(13): 7590–99.

Chen, Xiaojun et al. 2012. “Kaempferol Regulates MAPKs and NF-ΚB Signaling Pathways to Attenuate LPS-Induced Acute Lung Injury in Mice.” International Immunopharmacology 14(2): 209–16.

Calderón-Montaño, J M, E Burgos-Morón, C Pérez-Guerrero, and M López-Lázaro. 2011. “A Review on the Dietary Flavonoid Kaempferol | BenthamScience.” Mini reviews in medicinal chemistry 11(4): 298–344.

In the hamster, a chronic susceptible model to amoebic liver abscess, the reduction in the neutrophils’ MPO activity as well as reduction in ROS and NO production could be the mechanisms that could participate in the amoebic liver abscess resolution.

A= You are right

At the same time, for example, antioxidant enzymes, in particular E. histolytica peroxiredoxin, are known to play a role in the survival of trophozoites in the presence of ROS (see, for example, Sen, A., Chatterjee, N. S., Akbar, M. A., Nandi, N. , & Das, P. (2007) The 29-kilodalton thiol-dependent peroxidase of Entamoeba histolytica is a factor involved in pathogenesis and survival of the parasite during oxidative stress Eukaryotic Cell, 6(4), 664-673.).

It is possible that reducing the expression of antioxidant enzymes as with kaempferol will increase the sensitivity of E. histolytica to reactive oxygen species, but this will not increase the antiprotozoal activity of neutrophils, because they are also inactivated.

R= You are right, the kaempferol act directly on the Rr, Prx, TrxR and Trx antioxidant enzymes of the E. histolytica it has also reported that kaempferol has a directly effect on the amoebic cytoskeleton affecting its viability (Bolaños et al. 2015). Moreover, in a susceptible model of ALA the inflammatory cells lysis releases ROS, RNS, NO inducing a parenchyma damage, on the contrary, the kaempferol antioxidant and anti-inflammatory activity could inhibit the damage caused by the lysis of the inflammatory cells. 

Bolaños, Verónica et al. 2015. “Kaempferol Inhibits Entamoeba Histolytica Growth by Altering Cytoskeletal Functions.” Molecular and Biochemical Parasitology 204(1): 16–25.

In general, it's difficult.

Reviewer 2 Report

The MS studied in vitro evaluation of the antiamoebic activity of kaempferol, the overall quality of the MS is high, which also has a certain degree of innovation.

Minor issues;

1. Introduction part:It is recommended that the author add some highlights of this study in the introduction section. The introduction gives the impression that it is summarizing the research foundation of a large number of predecessors and does not provide any highlights for this study.

2. The highest concentration of kaempferol is 150 uM, is it too high? If it is converted into an oral dose, will it be too high.

3.  Flavonoids themselves have good biological activity, but their poor water solubility and low bioavailability seriously hinder their drug formation. Has the author considered the medicinal properties of this compound? Suggest adding content to the discussion section.

4. This manuscript mainly discusses the in vitro activity of kaempferol, but there is a lack of data on in vivo research. It is recommended that the author add in vivo research data if they have time.

Author Response

Reviewer 2.

Comments and Suggestions for Authors

The MS studied in vitro evaluation of the antiamoebic activity of kaempferol, the overall quality of the MS is high, which also has a certain degree of innovation.

Minor issues.

  1. Introduction partIt is recommended that the author add some highlights of this study in the introduction section. The introduction gives the impression that it is summarizing the research foundation of many predecessors and does not provide any highlights for this study.

R= We appreciate very much your review we believe that your comments enrich our work. You are right in your comment. Therefore, we add a new paragraph that highlighted our work (Lines 114-119) “interaction of amoebae with hamster neutrophils for short times demonstrating that kaempferol has a dual effect on amoeba and neutrophils. On the one hand, it diminished amoebic viability on the other hand kaempferol also decrease Neutrophil’s MPO activity and ROS and NO release. In a hamster susceptible model to amoebic liver abscess kaempferol could participate in the amoebic liver abscess resolution through its anti-inflammatory and antioxidant activities”.

  1. The highest concentration of kaempferol is 150 uM, is it too high? If it is converted into an oral dose, will it be too high.

R= As you can appreciate in figure 1, we standardized kaempferol concentration, 150 mM was the concentration where we observed the greatest effect. On the other hand, the same concentration was used on the cell line and had no harmful effect.

  1. Flavonoids themselves have good biological activity, but their poor water solubility and low bioavailability seriously hinder their drug formation. Has the author considered the medicinal properties of this compound? Suggest adding content to the discussion section.

R= There are several works, where kaempferol is used in in vivo models, where a regulatory effect was observed after treatment with this drug (Cao et al. 2014; Dabeek and Marra 2019; Tang et al. 2014). We add this at the introduction about in vivo effect of kaempferol (Lines 100-103) “scavenger [23]. Kaempferol has been reported to have an effect in suppressing ROS production in mouse bone marrow-derived neutrophils [24], moreover in a model of induced mouse mastitis [25], and a model of mouse model of LPS-induced acute lung injury treats with kaempferol, the MPO activation was reduced[26]”.

Cao, Rongfeng et al. 2014. “Protective Effects of kaempferol on Lipopolysaccharide-Induced Mastitis in Mice.” Inflammation 37(5): 1453–58.

Dabeek, Wijdan M., and Melissa Ventura Marra. 2019. “Dietary Quercetin and Kaempferol: Bioavailability and Potential Cardiovascular-Related Bioactivity in Humans.” Nutrients 11(10).

Tang, Xi Lan et al. 2014. “Protective Effect of Kaempferol on LPS plus ATP-Induced Inflammatory Response in Cardiac Fibroblasts.” Inflammation 38(1): 94–101.

  1. This manuscript mainly discusses the in vitro activity of kaempferol, but there is a lack of data on in vivo research. It is recommended that the author add in vivo research data if they have time.

R= Thank you for your comment, we added a briefly information in the introduction (Lines 100-103) “scavenger [23]. Kaempferol has been reported to have an effect in suppressing ROS production in mouse bone marrow-derived neutrophils [24], moreover in a model of induced mouse mastitis [25], and a model of mouse model of LPS-induced acute lung injury treats with kaempferol, the MPO activation was reduced[26]”.

Reviewer 3 Report

In this work the authors have investigated the antiamoebic effect of kaempferol. The authors have evaluated the effect of the drug directly on trophozoites of E. histolytica at the level of viability and gene expression of Pr, Rr and TrxR. They have also evaluated the effect on hamster neutrophils where they have observed a reduction in ROS, NO and MPO. It is an interesting work that proposes an alternative drug to the existing one. However, in my opinion, the results presented are not enough to support the conclusions of the authors and therefore, my recommendation is that it must not be accepted for publication in IJMS journal. Comments:

11. The effect of this compound could have a species-specific component. In this sense, they should explain why they have chosen the hamster as the representative animal model of the human.

22. The authors have evaluated oxidative stress molecules,however they should have considered evaluating the Th1/Th2 cytokine profile.

33. Why haven't the authors measured the expression of key mediators in the pathogenesis of E. histolytica such as proteases (essential for liver abscess formation), EhCP112, pore-forming proteins, or neurohormonal substances?

44. Regarding the expression of peroxiredixin, it is necessary to take into account two aspects. On the one hand, the authors have measured relative changes in gene expression. These changes may not be large enough in quantitative terms (the change is relative) and the RNA does not have necessary to be translated into proteins. An assessment of the amount of protein and enzymatic activity would be necessary to support the relative gene expression findings. On the other hand, not only the amount, but the location of these proteins is important. Lyer et al (Iyer LR, Singh N, Verma AK, Paul J. Differential expression and immunolocalization of antioxidant enzymes in Entamoeba histolytica isolates during metronidazole stress. Biomed Res Int. 2014;2014:704937) have shown how the location of Pr is essential to explain its protective role. Thus, it would be necessary to evaluate by some microscopic technique the intracellular localization of these proteins and if the drug has any effect on this.

55. Why do the authors assume that neutrophil dysfunction may be beneficial? It is true that E. histolytica can lyse neutrophils in vitro, causing the release of their contents, suggesting that the damage in liver abscesses is due to mediators released by neutrophils. However, larger lesions have been observed in neutropenic mice than in normal mice, what reinforces the role of neutrophils in resistance to infection (Velazquez C, Shibayama-Salas M, Aguirre-Garcia J, Tsutsumi V, Calderon J. Role of neutrophils in innate resistance to Entamoeba histolytica liver infection in mice. Parasite Immunol. 1998 Jun;20(6):255-62. doi: 10.1046/j.1365-3024.1998.00128.x. PMID: 9651927).

66. This work concludes that kaempferol is more effective than MTZ against E histolytica. However, studying the effect of this substance only on neutrophil activity is not enough. What happens with macrophages and eosinophils, key cells in the anti-parasitic response?

77. In this work, a histolytic E line is used. However, there are many differences in the effect of different drugs in relation to variations at the strain level. It would be convenient to include isolates from patients in order to verify the effects observed by the authors.

88. In the figure 1, the trophozoites viability changes are represented. Data represent the mean ± SD of n=3. How do the authors explain that the deviations represented are so small?. The same effect can be seen in figure 6.

99. Regarding protein overexpression data (figure 2), the authors found an increase of molecular bands of 67, 59, 27 or 24 kDa. It is not possible to resolve all the proteins in the electrophoresis bands. It would be necessary to resort to other techniques such as 2D electrophoresis.

T10. There is no mention to the n in the figure 4 legend. If it is n=3 and the mean and SD are represented, again the SDs seems to be very small.

111. The authors do not show photomicrographs of the isolated netrophils. It would be convenient to include them to estimate the purity and determine if they have been preactivated by the isolation process.  

112. Why have authors used VERO cells to test for cytotoxicity?

113. In relation to the primers used for gene expression experiments. Were they designed by the authors? IF so, how have they been validated? Have you verified that the method of the DDCt can be applied? How long are the amplicons?

Author Response

Reviewer 3.

Comments and Suggestions for Authors

In this work the authors have investigated the antiamoebic effect of kaempferol. The authors have evaluated the effect of the drug directly on trophozoites of E. histolytica at the level of viability and gene expression of Pr, Rr and TrxR. They have also evaluated the effect on hamster neutrophils where they have observed a reduction in ROS, NO and MPO. It is an interesting work that proposes an alternative drug to the existing one. However, in my opinion, the results presented are not enough to support the conclusions of the authors and therefore, my recommendation is that it must not be accepted for publication in IJMS journal. Comments:

  1. The effect of this compound could have a species-specific component. In this sense, they should explain why they have chosen the hamster as the representative animal model of the human.

R= We are very grateful to you for the careful reading of our manuscript. It is extensively documented since many years ago that the hamster model is considered a susceptible model of amoebic liver abscess. Therefore, this model has been used to research physiopathology of the amoebic liver lesion as well as in pharmacological studies of various drugs against amoebiasis ( Herrera-Martínez et al. 2022; Rangel-Castañeda et al. 2018; Pais-Morales et al. 2016; Carrero et al. 2006).

Herrera-Martínez, Mayra et al. 2022. “Alpha-Terthienyl Increases Filamentous Actin of Entamoeba Histolytica.” Molecular and Biochemical Parasitology 252: 111512.

Rangel-Castañeda, Itzia Azucena et al. 2018. “Amoebicidal Activity of Curcumin on Entamoeba Histolytica Trophozoites.” Journal of Pharmacy and Pharmacology 70(3): 426–33.

Pais-Morales, Jonnatan et al. 2016. “Resveratrol Induces Apoptosis-like Death and Prevents in Vitro and in Vivo Virulence of Entamoeba Histolytica.” PLoS ONE 11(1): 1–23.

Carrero, Julio C. et al. 2006. “Dehydroepiandrosterone Decreases While Cortisol Increases in Vitro Growth and Viability of Entamoeba Histolytica.” Microbes and Infection 8(2): 323–31.

  1. The authors have evaluated oxidative stress molecules; however, they should have considered evaluating the Th1/Th2 cytokine profile.

R = We appreciate your suggestion; however, our work is based on acute in in vitro model, therefore Th1/Th2 cells could be considered in a future chronic study

  1. Why haven't the authors measured the expression of key mediators in the pathogenesis of E. histolytica such as proteases (essential for liver abscess formation), EhCP112, pore-forming proteins, or neurohormonal substances?

R = This is an excellent suggestion; we will continue working in this model and in the future will consider measured other amoebic virulence molecules.

  1. Regarding the expression of peroxiredixin, it is necessary to take into account two aspects. On the one hand, the authors have measured relative changes in gene expression. These changes may not be large enough in quantitative terms (the change is relative) and the RNA does not have necessary to be translated into proteins. An assessment of the amount of protein and enzymatic activity would be necessary to support the relative gene expression findings. On the other hand, not only the amount, but the location of these proteins is important. Lyer et al (Iyer LR, Singh N, Verma AK, Paul J. Differential expression and immunolocalization of antioxidant enzymes in Entamoeba histolytica isolates during metronidazole stress. Biomed Res Int. 2014;2014:704937) have shown how the location of Pr is essential to explain its protective role. Thus, it would be necessary to evaluate by some microscopic technique the intracellular localization of these proteins and if the drug has any effect on this.

    R= Thank you for your comment, it’s very interesting. You are right in this study we demonstrated changes in RNA gene expression in future studies we will perform enzymatic assays with their cell location.

  1. Why do the authors assume that neutrophil dysfunction may be beneficial? It is true that E. histolytica can lyse neutrophils in vitro, causing the release of their contents, suggesting that the damage in liver abscesses is due to mediators released by neutrophils. However, larger lesions have been observed in neutropenic mice than in normal mice, what reinforces the role of neutrophils in resistance to infection (Velazquez C, Shibayama-Salas M, Aguirre-Garcia J, Tsutsumi V, Calderon J. Role of neutrophils in innate resistance to Entamoeba histolytica liver infection in mice. Parasite Immunol. 1998 Jun;20(6):255-62. doi: 10.1046/j.1365-3024.1998.00128.x. PMID: 9651927).

R= In our group we demonstrated that neutrophils are important against the amoeba damage in the amoebic liver abscess in mice however, in the susceptible model of ALA in hamster the exacerbated and uncontrolled immune response is characterized by the lysis of PMNs cells and the release of mediators that favor the increase of liver damage. The response to the amoeba is different in each model, in the case of the mouse model the neutrophils act efficiently against E. histolytica by eliminating them, therefore the mouse is considered a model of resistance (Contis Montes de Oca et al. 2020; Cruz-Baquero et al. 2017; Olivos-García 2007; Olivos-García et al. 2004; Jarillo-Luna, Campos-Rodriguez, and Tsutsumi 2000; Tsutsumi and Martinez-Palomo 1988; Tsutsumi et al. 1984).

Contis Montes de Oca, Arturo et al. 2020. “Neutrophil Extracellular Traps and MPO in Models of Susceptibility and Resistance against Entamoeba Histolytica.” Parasite Immunology 42(6): 1–12.

Cruz-Baquero, Andrea et al. 2017. “Different Behavior of Myeloperoxidase in Two Rodent Amoebic Liver Abscess Models.” PLoS ONE 12(8): 1–23.

Olivos-García, A. et al. 2007. “Late Experimental Amebic Liver Abscess in Hamster Is Inhibited by Cyclosporine and N-Acetylcysteine.” Experimental and Molecular Pathology 82(3): 310–15.

Olivos-García, A. et al. 2004. “Inflammation, Complement, Ischemia and Amoebic Survival in Acute Experimental Amoebic Liver Abscesses in Hamsters.” Experimental and Molecular Pathology 77(1): 66–71.

Jarillo-Luna, Rosa Adriana, Rafael Campos-Rodriguez, and Víctor Tsutsumi. 2000. “Morphological Changes of Mouse Liver Infected with Trophozoites of Entamoeba Histolytica.” Archives of Medical Research 31(4): S251–53.

Tsutsumi, V., and A. Martinez-Palomo. 1988. “Inflammatory Reaction in Experimental Hepatic Amebiasis. An Ultrastructural Study.” American Journal of Pathology 130(1): 112–19.

Tsutsumi, V., R. Mena-Lopez, F. Anaya-Velazquez, and A. Martinez-Palomo. 1984. “Cellular Bases of Experimental Amebic Liver Abscess Formation.” American Journal of Pathology 117(1): 81–91.

  1. This work concludes that kaempferol is more effective than MTZ against E histolytica. However, studying the effect of this substance only on neutrophil activity is not enough. What happens with macrophages and eosinophils, key cells in the anti-parasitic response?

R= You are right however our study is focus on acute response, the macrophage participates in the chronic stage of ALA, we will consider evaluating the role of kaempferol against macrophage and eosinophil in the future.

  1. In this work, a histolytic E line is used. However, there are many differences in the effect of different drugs in relation to variations at the strain level. It would be convenient to include isolates from patients in order to verify the effects observed by the authors.

R= Our work was mainly focused on evaluating the effect of kaempferol on a virulent amoeba strain that causes damage and has been extensively studied. Your observation is interesting and will be taken into account for future work. The virulence of E. histolytica was corroborated by the formation of amoebic liver abscesses in the susceptible model.

  1. In the figure 1, the trophozoites viability changes are represented. Data represent the mean ± SD of n=3. How do the authors explain that the deviations represented are so small? The same effect can be seen in figure 6.

R= The results presented are those we obtained with SD

  1. Regarding protein overexpression data (figure 2), the authors found an increase of molecular bands of 67, 59, 27 or 24 kDa. It is not possible to resolve all the proteins in the electrophoresis bands. It would be necessary to resort to other techniques such as 2D electrophoresis.

R= We observed with the SDS PAGE in one dimension the increase of different bands when amoebae were treated with kaempferol. Therefore, we can suggest the overexpression of some proteins. As you suggest we consider in the future to perform a 2D gel with mass spectrometry to define the exact proteins that are overexpressed.

T10. There is no mention to the n in the figure 4 legend. If it is n=3 and the mean and SD are represented, again the SDs seems to be very small.

R= Thank you for your observation we added in figure 4 the SD. The results presented are those we obtained with SD

  1. The authors do not show photomicrographs of the isolated neutrophils. It would be convenient to include them to estimate the purity and determine if they have been preactivated by the isolation process.

R= The technique for purifying neutrophils has already been described in previous work and is 95% purity ( Contis Montes de Oca et al. 2020; Behrendt et al. 2010; Reumaux et al. 2003). We add here a picture of neutrophils.

Contis Montes de Oca, Arturo et al. 2020. “Neutrophil Extracellular Traps and MPO in Models of Susceptibility and Resistance against Entamoeba histolytica.” Parasite Immunology 42(6): 1–12.

Behrendt, Jan Hillern et al. 2010. “Neutrophil Extracellular Trap Formation as Innate Immune Reactions against the Apicomplexan Parasite Eimeria Bovis.” Veterinary Immunology and Immunopathology 133(1): 1–8.

Reumaux, Dominique et al. 2003. “Expression of Myeloperoxidase (MPO) by Neutrophils Is Necessary for Their Activation by Anti-Neutrophil Cytoplasm Autoantibodies (ANCA) against MPO.” Journal of Leukocyte Biology 73(6): 841–49.

  1. Why have authors used VERO cells to test for cytotoxicity?

     R= Vero cells have been used for cytotoxicity assays ( Yuan et al. 2023; Chan and Hsu 2000; Muiva-Mutisya et al. 2018;). In our working group we usually use them to cytotoxic tests.

Yuan, Lu et al. 2023. “Mechanism for Inhibition of Cytotoxicity of Shiga Toxin by Luteolin.” Toxicology in Vitro 87: 105537.

Chan, P., and F-L. Hsu. 2000. “The in Vitro- Inhibitory Effect of Flavonoid Astilbin on 3 - Hydroxy-3-Methylglutaryl Coenzime A Reductase on VERO Cells.” Hearth,Lung and Circulation: 114.

Muiva-Mutisya, Lois M. et al. 2018. “Antiplasmodial Prenylated Flavanonols from Tephrosia Subtriflora.” Natural Product Research 32(12): 1407–14. https://doi.org/10.1080/14786419.2017.1353510.

  1. In relation to the primers used for gene expression experiments. Were they designed by the authors? IF so, how have they been validated? Have you verified that the method of the DDCt can be applied? How long are the amplicons?

R= Thank you for the observations, we review details and improve the section as follows:

“For the extraction of RNA from 15,000 E. histolytica trophozoites incubated with 150 µM of kaempferol or MTZ for 90 min at 37 °C, the TRIzol-Chloroform (Thermo Scientific, MA, USA) method was used. At the end of this process, the cells were resuspended in RNAse-free water. RNA was stored at -80 °C until use. RNA quantification was performed using a Nanodrop Lite (Thermo Scientific, Waltham, MA, USA). The isolated RNA was treated with RQ1 RNase-Free DNase (PROMEGA, WI, USA) to avoid genomic DNA contamination and cDNA was synthesized using the First Strand cDNA synthesis kit (Thermo Scientific, MA, USA) according to the manufacturer’s protocol. RT-qPCR was performed with a Step One Real-Time PCR system (Applied Biosystems, CA, USA) by monitoring the increase in fluorescence in real time using SYBR Green PCR Master Mix (Applied Biosystems, CA, USA). Melting curve protocols were performed to ensure the specificity of the amplification products. The primers were designed using Primer Express 3.0.1 software (Applied Biosystems, Foster City, CA, USA) and synthesized commercially (IDT Integrated DNA Technologies, Coralville, IA, USA) the size of all amplicons was designed of 150 pb. (Table 1). For E. histolytica trophozoites, primers specific for Rr, Prx, TrxR and Trx were used, as a control, Glyceraldehyde-3-phosphate dehydrogenase (GAPDH) from E. histolytica was used.

The application of the comparative Cycle Threshold (CT) method was conducted to validate the effect of treatment on the expression of two endogenous control (18S subunit ribosomal and glyceraldehyde-3-phosphate dehydrogenase (GAPDH). We selected GAPDH because not statistically significant relationship was found between the treatment and basal expression of this gene. The relative quantification of antioxidant enzymes was calculated using the CT method by applying the comparative cycle threshold CT, which uses the arithmetic formula 2−ΔΔCT [50]. To validate the method, we verified that the amplification efficiency for the target genes and the endogenous gene GAPDH, were nearly equal, examined CT variations with serial of cDNA template dilutions and a plot of log cDNA concentrations versus CT was made and efficiency was calculated using the equation E= -1+10(-1/slope). The statistical significance between the untreated and treated trophozoites was calculated using Bonferroni`s test with GraphPad Prism statistical software (GraphPad, San Diego, CA, USA)”.

Livak, Kenneth J., and Thomas D. Schmittgen. 2001. “Analysis of Relative Gene Expression Data Using Real-Time Quantitative PCR and the 2-ΔΔCT Method.” Methods 25(4): 402–8.

Round 2

Reviewer 1 Report

The reviewer is completely satisfied with the answers that the authors gave to his questions and comments; the manuscript can be accepted for publication in its present form.

Author Response

R= We are very grateful to your decision, if English language editing is recommended, we will request MDPI to review the English language editing.

Reviewer 3 Report

First, I want to sincerely thank the authors for their response to my comments. I want to emphasize that in my opinion it is an interesting job, with interesting results. Although the authors' comments have answered some of my objections, in general the major ones have not. It is only my opinion, but I consider that the results presented in this work cannot support the conclusions reached by the authors. Specifically, I continue to have doubts regarding the following aspects:

11. I still consider that it is important to evaluate the Th1/Th2 cytokine profile.

22. The expression of key mediators in the pathogenesis of E. histolytica such as proteases (essential for liver abscess formation), EhCP112, pore-forming proteins, or neurohormonal substances should be studied.

33. I believe that studies in relation to the location and activity of peroxiredixin are necessary to be able to establish conclusions in relation to this molecule.

44.  In my opinion, it is necessary to evaluate the effects on eosinophils and macrophages. Otherwise, it is possible to talk about the effects of this drug on isolated neutrophils, but not to extrapolate effects at other levels.

55. Questions regarding the interpretation of the effect on protein expression (fig 2) remain unanswered.

66. I still consider it necessary for the authors to show microphotographs of the isolated neutrophils. I believe that it is mandatory to demonstrate that they have not been activated or agglutinated by the isolation process and I do not believe that relying only on previous studies is enough.

77. Regarding the semi-comparative method used to analyze the PCR data, I would like to point out that it can only be used if the amplification efficiency is 1. Otherwise, artifacts are generated. For them it is necessary to make a cDNA curve. The amplification efficiency is higher the smaller the fragments to be amplified. 150 bp are borderline fragments, and therefore these primers need to be validated. Have the authors done this?

FFor all these reasons, I cannot change my initial recommendation. Once again, Ithank the authors for their courtesy in replying to my comments.

Author Response

First, I want to sincerely thank the authors for their response to my comments. I want to emphasize that in my opinion it is an interesting job, with interesting results. Although the authors' comments have answered some of my objections, in general the major ones have not. It is only my opinion, but I consider that the results presented in this work cannot support the conclusions reached by the authors. Specifically, I continue to have doubts regarding the following aspects:

R= We are very grateful for his thorough review of our manuscript. As we previously described in the present manuscript, we have analyzed the antiamoebic activity of kaempferol on E. histolytica trophozoites and in the interactions between amoebae and neutrophils. Multiple previous works have documented and demonstrated that neutrophils are the first cells that contact with E. histolytica at early times after the arrival of the amoeba to the hepatic parenchyma. These immune cells are the first line of defense against the parasite; however, depending on the animal model used, the effector response may or may not be effective. In the present work we used neutrophils from a model susceptible to amoebic liver abscess (ALA) such as the hamster. The response of the neutrophils is different between susceptible or resistant ALA model, when the neutrophils are properly activated eliminate the amoebae. In addition, if amoebae are not eliminated in the early stages of the infection, they trigger a prolongated and exaggerated inflammatory response that increase the amoebic damage. Studies on hamster model have determined that the inadequate immune response with the massive lysis of neutrophils exacerbates host tissue damage in ALA, this host tissue damage is caused more by the immune response than by the amoeba, here the acute inflammatory response is inefficient in destroying amoebae allowing the evolution of amoebic damage, a process aggravated by the lytic secreted products of the inflammatory cells. Since neutrophils are the first cells that contact Entamoeba histolytica at early stages of the inflammatory response, the aim of the current study is among others analyze the kaempferol activity in the interactions of amoebae and neutrophils from amoebic liver abscess-susceptible model such as hamster. Now we have added this highlights information in the introduction section of the manuscript, abstract line 25 and introduction lines 120-125, discussion lines 350-353, 386-390 (Olivos-García et al. 2007; Tsutsumi and Shibayama 2006; Perez-Tamayo et al. 2006; Olivos-García et al. 2004; Tsutsumi and Martinez-Palomo 1988; Tsutsumi et al. 1984; Herrera-Martínez et al. 2022; Rangel-Castañeda et al. 2018; Pais-Morales et al. 2016; Carrero et al. 2006).

Olivos-García, A. et al. 2007. “Late Experimental Amebic Liver Abscess in Hamster Is Inhibited by Cyclosporine and N-Acetylcysteine.” Experimental and Molecular Pathology 82(3): 310–15.

Tsutsumi, Víctor, and Mineko Shibayama. 2006. “Experimental Amebiasis: A Selected Review of Some in Vivo Models.” Archives of Medical Research 37(2): 210–20.

Perez-Tamayo, R. et al. 2006. "Patohgenesis of acute experimental liver amebiasis." Archives of Medical Research 37(2):203-209.

Olivos-García, A. et al. 2004. “Inflammation, Complement, Ischemia and Amoebic Survival in Acute Experimental Amoebic Liver Abscesses in Hamsters.” Experimental and Molecular Pathology77(1): 66–71.

Tsutsumi, V., and A. Martinez-Palomo. 1988. “Inflammatory Reaction in Experimental Hepatic Amebiasis. An Ultrastructural Study.” American Journal of Pathology 130(1): 112–19.

Tsutsumi, V., R. Mena-Lopez, F. Anaya-Velazquez, and A. Martinez-Palomo. 1984. “Cellular Bases of Experimental Amebic Liver Abscess Formation.” American Journal of Pathology 117(1): 81–91.

Herrera-Martínez, Mayra et al. 2022. “Alpha-Terthienyl Increases Filamentous Actin of Entamoeba Histolytica.” Molecular and Biochemical Parasitology 252: 111512.

Rangel-Castañeda, Itzia Azucena et al. 2018. “Amoebicidal Activity of Curcumin on Entamoeba HistolyticaTrophozoites.” Journal of Pharmacy and Pharmacology 70(3): 426–33.

Pais-Morales, Jonnatan et al. 2016. “Resveratrol Induces Apoptosis-like Death and Prevents in Vitro and in Vivo Virulence of Entamoeba Histolytica.” PLoS ONE 11(1): 1–23.

Carrero, Julio C. et al. 2006. “Dehydroepiandrosterone Decreases While Cortisol Increases in Vitro Growth and Viability of Entamoeba Histolytica.” Microbes and Infection 8(2): 323–31).

  1. I still consider that it is important to evaluate the Th1/Th2 cytokine profile.

R=Thank you for your comment. No doubt as you suggest the evaluation of the Th1/Th2 cytokine profile is very important however our study focuses mainly on short time interactions of amoebae and neutrophils. In the future it will be important to elucidate the mechanisms that occur with the chronic phase where cytokines and mononuclear cells are participating. Also, the special issue of the ijms is about neutrophils.

  1. The expression of key mediators in the pathogenesis of E. histolytica such as proteases (essential for liver abscess formation), EhCP112, pore-forming proteins, or neurohormonal substances should be studied.

R= We agree with your comment about the study of mediators in the pathogenesis of E. histolytica however these assays it will take a long to perform and our study is related to the short times of the interaction between neutrophils and amoebae in presence of Kaempferol. We will continue the analysis of other stages of the interaction among amoebic mediators and other immune cells.

  1. I believe that studies in relation to the location and activity of peroxiredixin are necessary to be able to establish conclusions in relation to this molecule.

R= In our study we made a first search for the amoebic targets on which Kaempferol acts. In the future we will continue working and exploring different amoebic targets.

  1. In my opinion, it is necessary to evaluate the effects on eosinophils and macrophages. Otherwise, it is possible to talk about the effects of this drug on isolated neutrophils, but not to extrapolate effects at other levels.

R= Thank you for your comment. Our study demonstrated the Kaempferol effect on interactions between hamster isolated neutrophils and amoebae, future in vivo studies on a susceptible model by our group with Kaempferol will be realized. Certainly. eosinophils and macrophage are very important in the host immune response however macrophages are relevant in the chronic stage of amoebic damage and our study is focused on short times where neutrophil participation is relevant.

  1. Questions regarding the interpretation of the effect on protein expression (fig 2) remain unanswered.

R= Thank you for your comment. As answered previously the SDS PAGE in one dimension showed the increase of different bands when amoebae were treated with kaempferol. Therefore, we can suggest the overexpression of some proteins. As you suggest we consider in the future to perform a 2D gel with mass spectrometry to define the exact proteins that are overexpressed. We are currently unable to perform this test.

  1. I still consider it necessary for the authors to show microphotographs of the isolated neutrophils. I believe that it is mandatory to demonstrate that they have not been activated or agglutinated by the isolation process and I do not believe that relying only on previous studies is enough.

R= Thank you for your suggestion. We now added a microphotograph of the isolate neutrophils has been showed as supplementary material (SI). Line 566.

  1. Regarding the semi-comparative method used to analyze the PCR data, I would like to point out that it can only be used if the amplification efficiency is 1. Otherwise, artifacts are generated. For them it is necessary to make a cDNA curve. The amplification efficiency is higher the smaller the fragments to be amplified. 150 bp are borderline fragments, and therefore these primers need to be validated. Have the authors done this?

R= Thank you very much for your valuable corrections, we showed details of quantitative PCR method used

The primers were designed using Primer Express 3.0.1 software (Applied Biosystems, Foster City, CA, USA). The size of all amplicons was designed of sizes less to 300 pb as indicated in quantitative PCR protocols (Applied Biosystems User Bulletin No. 2 (P/N 4303859, Livak and Livak and Schmittgen, 2001). The primers were synthesized commercially (IDT Integrated DNA Technologies, Coralville, IA, USA and amplicons sizes were of 150 pb with objective to obtain same amplifier efficiencies. Melting curve protocols were performed. To ensure the specificity of the amplification products (example figure 1). Moreover, size of all amplicons was verified in by electrophoresis on agarose gels.

Figure 1. Melting curve for the antioxidant gene E. histolytica Thioredoxin.

In addition, we verified that the amplification efficiency for the target genes and the endogenous gene, were nearly equal, examined CT variations with serial of cDNA template dilutions and a plot of log cDNA concentrations versus CT was made and efficiency was calculated using the equation Efficiency E= 10 (-1/slope)-1 (Rasmussen 2001). An example is showed in the figure 2.

E= 1.048012029

Figure 2. Determination of the amplification efficiency for the antioxidant gene Thioredoxin.

References

  1. Applied Biosystems User Bulletin No. 2 (P/N 4303859)
  2. Livak KJ, Schmittgen TD: Analysis of relative gene expression data using real-time quantitative PCR and the 2(-Delta Delta CT) Method. Methods 2001, 25:402-408.
  3. Rasmussen RP: Quantification on the In S. Meuer, C.T. Wittwer, and K Nakagawara(Eds.), Rapid Cycle Real-time PCR, Methods and Applications. Springer Press, Heidelberg 2001, 21-34.

For all these reasons, I cannot change my initial recommendation. Once again, Ithank the authors for their courtesy in replying to my comments.

Submission Date

13 May 2023

Date of this review

13 Jun 2023 18:23:12

Round 3

Reviewer 3 Report

Again, I want to sincerely thank the authors of this manuscript for their response to my comments. I want to highlight my most sincere respect for the research carried out and I want to express my respect for the findings shown by the authors. In general, I consider that they have responded to most of my comments, clarifying the doubts that arose the first time I read the manuscript. My opinion regarding the publication of this work, after the two rounds of comments has changed and after considering the changes introduced in it, I think it should be accepted for publication. Congratulations to the authors. I only have three small changes that in my opinion should be considered in the final version of this recording, but I leave this to the editor's consideration:

1.      Regarding the importance of the location and activity of peroxiredixin, it should be taken in consideration, at least in the discussion of the manuscript. I think it is important for readers to consider that in future research it is necessary to study this parameter.

2.      Concerning the effects on protein expression, the limitation of the technique used must be discussed in the manuscript.

3.      In relation to the photographs of the isolated neutrophils presented as supplementary material, they only show the nuclei of the leukocytes. It would be useful to show a Giemsa stain to verify that the isolated neutrophils have a normal morphology and that they have not been activated during the isolation process, something that can occur when working with these cells.

Finally, I want to thank again the authors for the kind and polite tone in their response to my comments and thank them for their patience in this process.

Author Response

Reviewer 3

Again, I want to sincerely thank the authors of this manuscript for their response to my comments. I want to highlight my most sincere respect for the research carried out and I want to express my respect for the findings shown by the authors. In general, I consider that they have responded to most of my comments, clarifying the doubts that arose the first time I read the manuscript. My opinion regarding the publication of this work, after the two rounds of comments has changed and after considering the changes introduced in it, I think it should be accepted for publication. Congratulations to the authors. I only have three small changes that in my opinion should be considered in the final version of this recording, but I leave this to the editor's consideration:

R= We greatly appreciate your comments and observations, which have undoubtedly enriched our work.

  1. Regarding the importance of the location and activity of peroxiredixin, it should be taken in consideration, at least in the discussion of the manuscript. I think it is important for readers to consider that in future research it is necessary to study this parameter.

R= Thank for you observation, we have added in discussion section this consideration respects the distribution and activities of the antioxidant enzymes. Lines 289-290.

  1. Concerning the effects on protein expression, the limitation of the technique used must be discussed in the manuscript.

R= Thank you very much for your comment, now we have added in discussion the limitations of this technique and the test that can be performed in the future. Lines 271-276.

  1. In relation to the photographs of the isolated neutrophils presented as supplementary material, they only show the nuclei of the leukocytes. It would be useful to show a Giemsa stain to verify that the isolated neutrophils have a normal morphology and that they have not been activated during the isolation process, something that can occur when working with these cells.

R= As has been extensively demonstrated in publications by us and in previous studies, neutrophil activation is observed by the release of extracellular DNA (NETosis) upon any foreign stimulus. It has been observed that neutrophils when isolated and not stimulated, do not show release of extracellular DNA [4,5]. In our supplementary confocal picture, we show neutrophils that do not show the release of extracellular DNA labeled by Sytox Green, which indicates that there is no activation.

  1. Contis Montes de Oca, A.; Cruz Baquero, A.; Campos Rodríguez, R.; Cárdenas Jaramillo, L.M.; Aguayo Flores, J.E.; Rojas Hernández, S.; Olivos García, A.; Pacheco Yepez, J. Neutrophil Extracellular Traps and MPO in Models of Susceptibility and Resistance against Entamoeba Histolytica. Parasite Immunol. 2020, 42, 1–12, doi:10.1111/pim.12714.
  2. Contis-Montes de Oca, A.; Carrasco-Yépez, M.; Campos-Rodríguez, R.; Pacheco-Yépez, J.; Bonilla-Lemus, P.; Pérez-López, J.; Rojas-Hernández, S. Neutrophils Extracellular Traps Damage Naegleria Fowleri Trophozoites Opsonized with Human IgG. Parasite Immunol. 2016, 38, 481–495, doi:10.1111/pim.12337.
  3. Klink, M.; Bednarska, K.; Blus, E.; Kielbik, M.; Sulowska, Z. Seasonal Changes in Activities of Human Neutrophils in Vitro. Inflamm. Res. 2012, 61, 11–16, doi:10.1007/s00011-011-0382-x.
  4. Sabbatini, M.; Bona, E.; Novello, G.; Migliario, M.; Renò, F. Aging Hampers Neutrophil Extracellular Traps (NETs) Efficacy. Aging Clin. Exp. Res. 2022, 34, 2345–2353, doi:10.1007/s40520-022-02201-0.
  5. Murata, H.; Kinoshita, M.; Yasumizu, Y.; Motooka, D.; Beppu, S.; Shiraishi, N.; Sugiyama, Y.; Kihara, K.; Tada, S.; Koda, T.; et al. Cell-Free DNA Derived From Neutrophils Triggers Type 1 Interferon Signature in Neuromyelitis Optica Spectrum Disorder. Neurol. Neuroimmunol. neuroinflammation 2022, 9, 1–12, doi:10.1212/NXI.0000000000001149.

Finally, I want to thank again the authors for the kind and polite tone in their response to my comments and thank them for their patience in this process.
